# Boundary Embedding Shaping with Adaptive Contrastive Learning for Graph Structural Disentanglement

Jiaqing Chen [* 1]   Zidu Yin [* 1]   Yichao Cai [2]   Yuhang Liu [2]   Zhen Zhang [2]   Dong Gong [3]   Javen Qinfeng Shi [2]

## Abstract

Graph neural networks (GNNs) excel at aggregating neighbor information for classification, yet their performance is hindered by graph structural entanglement, where spurious correlations from semantically irrelevant neighbors contaminate node embeddings. This challenge is most acute for nodes near class boundaries in the embedding space, where amplified structural noise blurs decision boundaries and destabilizes predictions. Existing robust GNN methods largely treat all nodes uniformly, ignoring boundary vulnerabilities. In this paper, to improve classification performance, we tackle graph structural disentanglement by identifying boundary-region entanglement as the primary bottleneck and propose **B**oundary **E**mbedding **S**haping (BES)[1], an adaptive contrastive learning GNN plug-in module that selectively suppresses spurious structural noise at decision boundaries with minimal model parameter perturbation. Extensive experiments demonstrate that BES consistently improves boundary discrimination and outperforms existing leading methods. Notably, BES boosts GCN performance by an average of 3.3% in node classification (up to 5.0% on WikiCS) and achieves superior accuracy in link prediction.

---

[*]Equal contribution [1]School of Information Science and Technology, Yunnan Normal University, Kunming, China [2]Australian Institute for Machine Learning, Adelaide University, Adelaide, Australia [3]School of Computer Science and Engineering, The University of New South Wales, Sydney, Australia. Correspondence to: Zidu Yin <zidu.yin@ynnu.edu.cn>.

*Proceedings of the $43^{rd}$ International Conference on Machine Learning*, Seoul, South Korea. PMLR 306, 2026. Copyright 2026 by the author(s).

[1]Code is available at https://github.com/coodest/BES.

## 1. Introduction

Graph data, characterized by intricate topologies and rich semantic interdependencies, underpins diverse applications spanning social networks (Lyu et al., 2024; Subramonian et al., 2024), biological systems (Zhang et al., 2025), computational chemistry (Li et al., 2025), and recommendation systems (Zhang et al., 2024b; Wu et al., 2022). Graph neural networks (GNNs) have emerged as the dominant paradigm for learning on such data by aggregating information from local neighborhoods to construct node embeddings. However, unlike grid-structured modalities such as images (Deng, 2012; Deng et al., 2009), graphs exhibit irregular topologies where neighborhood structures vary significantly across nodes, introducing noise and irrelevant information into node embeddings, particularly in classification. This leads to spurious correlations, a phenomenon referred to as graph structural entanglement (Zhu et al., 2020; Rusch et al., 2023), which blurs the decision boundaries between categories and renders classification notably more challenging.

Consider computer science literature classification from Wikipedia (Mernyei & Cangea, 2020) (Figure 1 (a)), where articles on *Internet protocols* and *computer security* exhibit substantial thematic overlap due to their inherent interconnection, i.e., protocols ensure delivery while security guarantees confidentiality. Distinguishing primary categories becomes challenging as articles often discuss techniques common to both domains.

In this work, we improve graph classification performance through graph structural disentanglement. We investigate how spurious structural noise from entangled neighborhoods degrades accuracy at decision boundaries and propose to refine boundary embeddings where structural disentanglement yields maximum classification gains. To this end, we propose **B**oundary **E**mbedding **S**haping (BES), an end-to-end framework that immunizes GNNs against spurious correlations from entangled neighbors by employing boundary-focused contrastive learning (You et al., 2020; Zhu et al., 2021) with minimal model parameter perturbations.

Specifically, BES builds upon existing GNN encoders to obtain comprehensive initial representations. We then employ

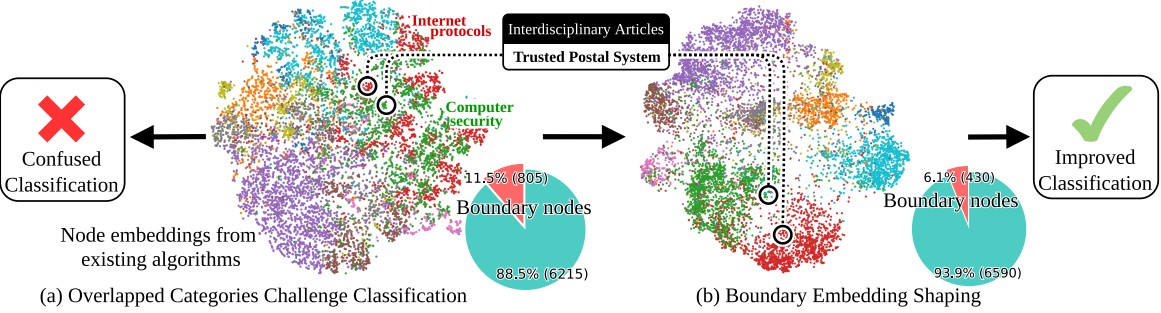

*Figure 1.* Boundary embedding shaping on WikiCS. BES structurally disentangles boundary nodes between clusters, producing sharply separable embeddings that significantly improve classification performance.

hard example mining (Shrivastava et al., 2016; Xia et al., 2022) to identify boundary nodes, namely nodes predominantly surrounded by neighbors from different classes. Subsequently, we selectively improve only those nodes whose embeddings can be separated with minimal parameter updates (Koh & Liang, 2017), allowing the model to sharpen boundaries without affecting non-boundary nodes. Finally, a GNN-based decoder performs classification using the improved embeddings. As shown in Figure 1(b), BES effectively filters spurious structural noise, producing sharply improved decision boundaries for robust classification.

In summary, our contributions are as follows:

- We identify spurious structural noise from entangled neighborhoods as a fundamental bottleneck in graph classification and provide theoretical analysis demonstrating how boundary-focused contrastive learning provably disentangles invariant factors from spurious correlations.

- We propose Boundary Embedding Shaping (BES), a principled framework that combines hard example mining with adaptive contrastive learning to selectively suppress structural noise at decision boundaries while preserving model stability through controlled parameter updates.

- Extensive experiments on node classification and link prediction benchmarks show that BES consistently outperforms existing leading methods, improving GCN accuracy by up to 5.0% on WikiCS, with observable gains from boundary nodes, validating our theoretical insights.

## 2. Preliminaries

**Graph Notation.** A graph $\mathcal{G} = (\mathcal{V}, \mathcal{E}, \mathbf{X})$ consists of a node set $\mathcal{V}$ with $N = |\mathcal{V}|$ nodes, an edge set $\mathcal{E} \subseteq \mathcal{V} \times \mathcal{V}$, and node feature matrix $\mathbf{X} \in \mathbb{R}^{N \times D}$, where $D$ denotes the feature dimensionality.

**Graph Encoder.** A graph encoder $\Phi$ learns node embeddings $\hat{\mathbf{z}}$ by aggregating neighbor information from $\mathbf{X}$ through message passing. However, this aggregation introduces structural entanglement (Zhu et al., 2020; Rusch et al., 2023), where spurious correlations from irrelevant neighbors contaminate node embeddings and blur decision boundaries, particularly for nodes near class boundaries.

**Decoder for Prediction.** For downstream tasks, a decoder $\Psi$ maps learned node embeddings to predictions. For node classification, given the embedding $\hat{\mathbf{z}}_i \in \hat{\mathbf{z}}$ of node $v_i \in \mathcal{V}$, the predicted label is:

$$\hat{y}_i = \Psi\left(\hat{\mathbf{z}}_i\right). \tag{1}$$

For edge classification, given embeddings of nodes $v_i$ and $v_j$ connected by an edge $e_{ij} \in \mathcal{E}$, the predicted edge label is:

$$\hat{y}_{ij} = \Psi\left(\hat{\mathbf{z}}_i, \hat{\mathbf{z}}_j\right). \tag{2}$$

## 3. Structural Disentanglement Theory

Our approach improves classification performance through principled graph structural disentanglement. To formalize this, we first introduce a latent variable model that captures the underlying latent data generation process, and present the corresponding technical assumptions in Section 3.1. This foundation enables a more principled approach to contrastive embedding refinement. Subsequently, we establish theoretical identifiability results in Section 3.2, with properties in Section 3.3.

### 3.1. Latent Variable Model for the Data Generative Process

Given the individual observational graph node feature variable $\mathbf{x} \in \mathcal{X}$, where $\mathcal{X} \subseteq \mathbb{R}^D$ denotes the data manifold of the observations, our goal is to extract class-determining information that remains invariant across diverse local neighboring structures. To facilitate theoretical analysis, we model the underlying data generation process using a la-

tent variable model (LVM) (Bishop, 1998).

Formally, let $\mathbf{z} \in \mathcal{Z} \subseteq \mathbb{R}^n$ be a continuous random variable within an open and simply connected *latent space* $\mathcal{Z}$. We decompose the latent space $\mathcal{Z}$ into two simply connected open subspaces, denoted as $\mathcal{Z} = \mathcal{Z}_{inv} \times \mathcal{Z}_{var}$. The latent variables in $\mathcal{Z}$ are represented as $\mathbf{z} = (\mathbf{z}_{inv}, \mathbf{z}_{var})$, where:

- $\mathbf{z}_{inv} \in \mathcal{Z}_{inv} \subseteq \mathbb{R}^{n_{inv}}$ denotes the *invariant latent variables*, capturing stable node information from relevant local neighboring structural properties (e.g., representative categorical characteristics) that remain consistent across different graph observations of the same class, thereby determining the class label $y$.

- $\mathbf{z}_{var} \in \mathcal{Z}_{var} \subseteq \mathbb{R}^{n-n_{inv}}$ denotes the *variant latent variables*, capturing dynamic relationships (e.g., temporary connectivity with outside current category) that vary across local neighboring structures, which is the noise source.

We allow a statistical relationship between $\mathbf{z}_{inv}$ and $\mathbf{z}_{var}$ without explicitly modeling the causal relationship, except in the case where $\mathbf{z}_{var} \to \mathbf{z}_{inv}$. In this scenario, changes in the local neighboring structure affecting $\mathbf{z}_{var}$ would inevitably propagate to $\mathbf{z}_{inv}$, which contradicts the desired invariance.

Then, let $\mathbf{g} : \mathcal{Z} \to \mathcal{X}$ be a smooth and invertible mapping with a smooth inverse (i.e., a diffeomorphism) to the observation space $\mathcal{X} \subseteq \mathbb{R}^d$, and let $\mathbf{x}$ denote the continuous random variable representing the observed node features. The generative process for the dataset of observational features $\mathbf{x}$ is then given by:

$$\mathbf{z} = (\mathbf{z}_{inv}, \mathbf{z}_{var}) \sim p_{\mathbf{z}}, \quad y = g_y(\mathbf{z}_{inv}), \quad \mathbf{x} = \mathbf{g}(\mathbf{z}), \quad (3)$$

where $p_{\mathbf{z}}$ is a regular density that is positive almost everywhere over $\mathcal{Z}$, and $g_y$ is a deterministic function indicating that $\mathbf{z}_{inv}$ exclusively determines the class label $y$.

Next, we formalize the construction of positive pairs for contrastive learning. Similar to the previous setting in image data (von Kügelgen et al., 2021), we connect the two nodes from different local sub-graphs in a positive pair, i.e., $\mathbf{x} = \mathbf{g}(\mathbf{z})$ and $\tilde{\mathbf{x}} = \mathbf{g}(\tilde{\mathbf{z}})$ where $\tilde{\mathbf{z}} = (\tilde{\mathbf{z}}_{inv}, \tilde{\mathbf{z}}_{var})$, through a conditional distribution $p_{\tilde{\mathbf{z}}|\mathbf{z}}$. Formally, we introduce the following two assumptions:

**Assumption 3.1** (Consistent Invariant Variables). *Assume that the positive pair construction process is label-informed, i.e., the two nodes from different local sub-graphs $\mathbf{x}$ and $\tilde{\mathbf{x}}$ in a positive pair share the same class label $y$. Then, the conditional density $p_{\tilde{\mathbf{z}}|\mathbf{z}}$ over $\mathcal{Z} \times \mathcal{Z}$ takes the form*

$$p_{\tilde{\mathbf{z}}|\mathbf{z}}(\tilde{\mathbf{z}}|\mathbf{z}) = \delta(\tilde{\mathbf{z}}_{inv} - \mathbf{z}_{inv}) \, p_{\tilde{\mathbf{z}}_{var}|\mathbf{z}_{var}}(\tilde{\mathbf{z}}_{var}|\mathbf{z}_{var})$$

*for some continuous conditional density $p_{\tilde{\mathbf{z}}_{var}|\mathbf{z}_{var}}$ on $\mathcal{Z}_{var} \times \mathcal{Z}_{var}$, where $\delta(\cdot)$ is the Dirac delta function, indicating that $\tilde{\mathbf{z}}_{inv} = \mathbf{z}_{inv}$ almost everywhere.*

**Assumption 3.2** (Sufficient Changes in Variant Latent Variables). *The conditional density $p_{\tilde{\mathbf{z}}_{var}|\mathbf{z}_{var}}$ stated in Asm. 3.1 is smooth w.r.t. both $\tilde{\mathbf{z}}_{var}$ and $\mathbf{z}_{var}$. Moreover, for any $\mathbf{z}_{var}$, the density satisfies $p_{\tilde{\mathbf{z}}_{var}|\mathbf{z}_{var}}(\cdot|\mathbf{z}_{var}) > 0$ within some open, non-empty subspace $\mathcal{O}(\mathbf{z}_{var}) \subseteq \mathcal{Z}_{var}$ containing $\mathbf{z}_{var}$.*

*Remark* 3.1. Asm. 3.2 requires only *local* support of the conditional density around each conditioning node, which is considerably weaker than assuming global coverage over the entire variant latent space $\mathcal{Z}_{var}$. This formulation aligns with standard assumptions in contrastive learning identifiability theory (von Kügelgen et al., 2021; Zimmermann et al., 2021). Crucially, this aligns perfectly with the inherent data characteristics of graph structures, especially in *heterophilic graphs*. In such settings, positive sample pairs sharing the same core semantic class ($\mathbf{z}_{inv}$) frequently connect to diverse and structurally different local neighborhoods, causing variations across the graph topology ($\mathbf{z}_{var}$). Our method turns this heterophily characteristic into an advantage: it natively fulfills the *sufficient variation* mandate by directly mining contrastive boundary pairs from data with high structural variation (detailed in Section 4), rather than relying on explicit artificial data augmentation.

These assumptions ensure that the positive pair construction preserves class consistency while allowing sufficient changes in the variant latent variables, thus facilitating robust contrastive learning.

### 3.2. Identifiability Analysis

Based on the proposed LVM in Section 3.1, our objective is to disentangle the block of invariant latent variables. To this end, we restate the definition of block-identifiability in Defn. 3.1 to align with the notations used in our setting, as this formulation is widely adopted for theoretical analysis in contrastive learning (von Kügelgen et al., 2021; Yao et al., 2024; Cai et al., 2025):

**Definition 3.1** (Block-Identifiability). We say that the true invariant latent variables $\mathbf{z}_{inv} = \mathbf{g}^{-1}(\mathbf{x})_{1:n_{inv}}$ are block-identifiable by a function $\mathbf{f} : \mathcal{X} \to \mathcal{Z}$ if the inferred representations $\hat{\mathbf{z}} = \mathbf{f}(\mathbf{x})$ contain all and only the information about $\mathbf{z}_{inv}$. In other words, there exists an invertible function $\mathbf{r} : \mathbb{R}^{n_{inv}} \to \mathbb{R}^{n_{inv}}$ between the true invariant latent variables and the learned representations, such that $\hat{\mathbf{z}} = \mathbf{r}(\mathbf{z}_{inv})$.

We then present the following identifiability result (a detailed proof can be found in Appendix A.1):

**Theorem 3.1** (Isolating Invariant Latent Variables). *Let $(\mathbf{x}, \tilde{\mathbf{x}})$ be a positive pair of observations generated accord-*

ing to the LVM formalized in Section 3.1, and suppose the positive pair construction process follows Asms. 3.1 and 3.2. Let $\mathbf{f} : \mathcal{X} \rightarrow (0,1)^{n_{inv}}$ be a sufficiently flexible and smooth function, where $(0,1)^{n_{inv}}$ represents a Cartesian product of open intervals in $n_{inv}$-dimensional space. Then, minimizing the following contrastive loss function with the temperature parameter set to $\tau = 1$ (similar to SimCLR (Chen et al., 2020)) over a sufficient number of sample pairs $(\mathbf{x}, \tilde{\mathbf{x}})$ ensures that $\mathbf{f}$ block-identifies $\mathbf{z}_{inv}$:

$$\mathcal{L}(\mathbf{z}, \tilde{\mathbf{z}}) = -\log \frac{\exp\left(\text{sim}^{pos}(\mathbf{z}, \tilde{\mathbf{z}})/\tau\right)}{\sum_{\mathbf{z}' \in Z, \mathbf{z}' \neq \mathbf{z}} \exp\left(\text{sim}^{neg}(\mathbf{z}, \mathbf{z}')/\tau\right)}$$

where $\text{sim}(\cdot, \cdot)$ is a similarity function. Therefore, the learned node embeddings include all and only the information of the invariant latent variables $\mathbf{z}_{inv}$.

**Discussion.** The invariant latent variables $\mathbf{z}_{inv} \in \mathcal{Z}_{inv} \subseteq \mathbb{R}^{n_{inv}}$ represent stable structural properties (e.g., community structure or node rules) that remain consistent across graph observations of the same class, thereby determining the class label $y$. The identifiability result stated above ensures that the learned representations effectively isolate and preserve the invariant latent components while discarding variant information, thereby enhancing classification performance. The loss in Thm. 3.1 is important to effectively filter the noise information from the local topology, which will be discussed in Section 4.

### 3.3. Properties of the Invariant Latent Variables Isolation

To further characterize the quality of the learned node embeddings and guide our specific design of embedding shaping, we derive the following properties of the invariant latent variables isolation for structure-level disentanglement.

(1) Cor. 3.1 justifies our objective of learning invariant features (a detailed proof can be found in Appendix A.2).

**Corollary 3.1** (Structure-Level Disentanglement). *Minimizing the contrastive loss under Asms. 3.1 and 3.2 maximizes the mutual information between the node embeddings $\hat{\mathbf{z}}$ and the invariant structure $\mathbf{z}_{inv}$, while minimizing information about the variant topology $\mathbf{z}_{var}$:*

$$\max_{\mathbf{f}} I(\hat{\mathbf{z}}; \mathbf{z}_{inv}) \quad s.t. \quad I(\hat{\mathbf{z}}; \mathbf{z}_{var}) \rightarrow 0.$$

*This confirms that the optimal node embeddings achieve structure-level disentanglement, effectively filtering structural noise.*

(2) Prop. 3.1 reveals that the disentanglement error is inversely proportional to the boundary margin (a detailed proof can be found in Appendix A.3).

**Proposition 3.1** (Disentanglement Error Bound). *Let $\epsilon = \mathbb{E}[\mathcal{L}]$ be the expected loss. The mean squared error between*

the learned representation $\hat{\mathbf{z}}$ and the true invariant factors (up to linear transformation $A$) is bounded by:

$$\inf_{A} \mathbb{E}[\|\hat{\mathbf{z}} - A\mathbf{z}_{inv}\|^2] \leq \mathcal{O}(\sqrt{\epsilon}) + \mathcal{O}\left(\frac{1}{margin(\partial \mathcal{Z}_{inv})}\right),$$

*where $margin(\partial \mathcal{Z}_{inv})$ denotes the separation margin at the class boundaries in the embedding space.*

*Remark* 3.2. Based on the error bound in Prop. 3.1, dynamic learning rate scheduling can be employed to balance the disentanglement error and the boundary margin during iterative optimization, leading to more stable disentanglement of local graph structures, as described in previous works (e.g., (Ma et al., 2019; Wang et al., 2020)).

**Theoretical Motivation for Practice.** Cor. 3.1 justifies our objective of learning invariant features. Crucially, Prop. 3.1 reveals that the disentanglement error is inversely proportional to the boundary margin. This theoretical insight directly motivates our **Boundary Embedding Shaping**, which explicitly aims to increase the margin for "hard" boundary nodes where the error bound is loosest.

## 4. Boundary Embedding Shaping

Building on the theoretical insights developed in Section 3, we now present the boundary embedding shaping framework to improve the embeddings from existing GNN encoders by training the attention layer $\mathcal{A}^{(B)}$, shown in Figure 2. BES consists of (1) the pairwise objective that instantiates the similarity function in Thm. 3.1 for the structural disentanglement (Section 4.1), (2) a gradient-equivalent center-based approximation of the pairwise objective (Section 4.2), and (3) an adaptive learning mechanism motivated by Prop. 3.1 that controls model parameter updates effectively (Section 4.3).

The initial node embeddings $\Phi(\mathbf{X})$ from existing GNN encoders, denoted as $\Phi = \{\Phi_1, \cdots, \Phi_k\}$, are used to obtain the improved node embeddings during boundary embedding shaping. See Appendix C.1 for more details.

### 4.1. Pairwise Contrastive Objective

To meet the condition in Asm. 3.2 and guarantee the disentanglement of the invariant topological structure, we first identify nodes with high structural variability. This variability can be empirically indicated by a normalized shift score $S$ based on local neighbors:

$$S(v_i) = \frac{1}{|\mathcal{N}_{\text{emb}}(v_i)|} |\{v_j \in \mathcal{N}_{\text{emb}}(v_i) : y_i \neq y_j\}|, \quad (4)$$
$$\forall v_i \in \mathcal{B},$$

where $\mathcal{B}$ is the node set for the boundary region, $\mathcal{N}_{\text{emb}}(v_i)$ denotes the neighbors of $v_i$ in the embedding space (e.g.,

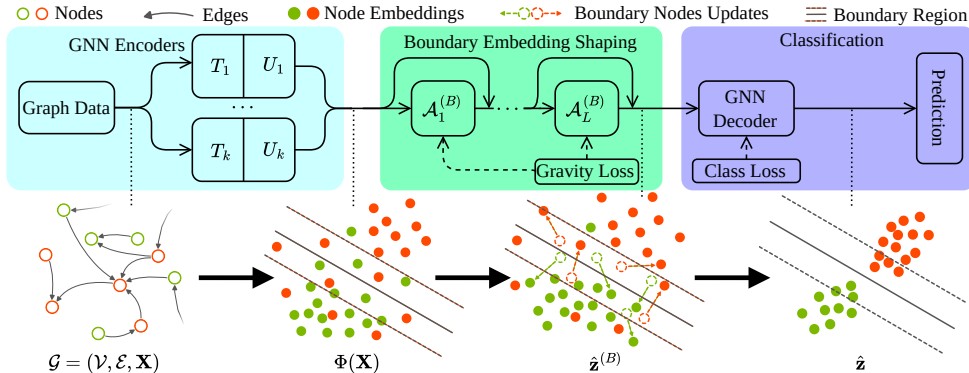

*Figure 2.* Boundary embedding shaping architecture aims to filter the irrelevant noise introduced by the variability of the local neighboring topology to obtain improved node embeddings for better classification.

$k$-nearest neighbors), and $y_i$ is the category label of node $v_i$. Instead of evaluating shift scores globally across the graph, we first coarsely mark the boundary region $\mathcal{B}$ between clusters $m$ and $n$ using a margin-based slab around the separating direction of the cluster centers, as defined in Equation (5).

$$\mathcal{B} := \left\{ \mathbf{x_i} \in \mathbf{X} \mid \left|(\boldsymbol{\mu}_m - \boldsymbol{\mu}_n)^\top \boldsymbol{\Sigma}^{-1} \Phi(\mathbf{x_i})\right| \leq \delta \right\}, \quad (5)$$

which denotes the boundary region between two node embedding clusters, each characterized by a Gaussian distribution $\mathcal{N}(\boldsymbol{\mu}_m, \boldsymbol{\Sigma}_m)$. We use embedding space neighbors within the boundary region rather than those from the original graph structure, as their features contribute more directly to the learning objective. Furthermore, our empirical evaluations demonstrate that the embedding space estimation similarly preserves the heterophilic characteristics from the original graph, while providing a significantly more stable and effective signal for downstream tasks (see Appendix D.1). Nodes within the boundary region exhibiting high structural variability (e.g., $S(v_i) > 0.5$) are designated as boundary nodes $v_b$, as they possess the maximal diversity of observations needed for structural disentanglement. Crucially, the identification explicitly leverages label information solely from training nodes, while strictly excluding all test nodes and test labels.

For a designated boundary node $v_b$, Thm. 3.1 dictates that minimizing the contrastive loss with appropriately defined similarity functions $\text{sim}^{pos}$ and $\text{sim}^{neg}$ is required. Specifically, the embedding $\Phi(\mathbf{x_b})$ should be aligned with all positive samples $v_p$ sharing the same label $y_b$ (pulling with positive gravity), while being separated from negative samples $v_n$ of different classes (pushing with negative gravity). Based on squared Euclidean distance, the pairwise gravity similarity terms are defined as:

$$\text{sim}_g^{pos}(\Phi(\mathbf{x_b}), \Phi(\mathbf{x}_p)) = -\|\Phi(\mathbf{x_b}) - \Phi(\mathbf{x}_p)\|_2^2, \\ v_p \in \mathcal{C}_{y_b}, \quad (6)$$

$$\text{sim}_g^{neg}(\Phi(\mathbf{x_b}), \Phi(\mathbf{x}_n)) = -\|\Phi(\mathbf{x_b}) - \Phi(\mathbf{x}_n)\|_2^2, \\ v_n \notin \mathcal{C}_{y_b}, \quad (7)$$

where $\mathcal{C}_{y_b}$ denotes the subset of nodes belonging to the same category as $v_b$. These pairwise gravity similarity terms directly instantiate the $\text{sim}(\cdot, \cdot)$ in the contrastive loss of Thm. 3.1, inheriting all its identifiability and disentanglement properties, and we call the overall contrastive objective the gravity loss for structural disentanglement.

### 4.2. Efficient Approximation of the Pairwise Objective

While the exact pairwise $\text{sim}_g^{pos}$ and $\text{sim}_g^{neg}$ formulation mathematically inherits the identifiability and properties claimed in Section 3, enumerating all pairwise similarities alongside global shift score evaluations requires $\mathcal{O}(N^2)$ complexity, which is prohibitive for massive graphs. However, observe that for the pairwise similarity aggregation, taking $\text{sim}_g^{pos}$ for example, for any class $\mathcal{C}_{y_b}$, we have the following geometric identity:

$$\sum_{v_p \in \mathcal{C}_{y_b}} \|\Phi(\mathbf{x_b}) - \Phi(\mathbf{x}_p)\|_2^2 = |\mathcal{C}_{y_b}| \cdot \|\Phi(\mathbf{x_b}) - \boldsymbol{\mu}_b\|_2^2 \\ + \sum_{v_p \in \mathcal{C}_{y_b}} \|\Phi(\mathbf{x}_p) - \boldsymbol{\mu}_b\|_2^2, \quad (8)$$

where $\boldsymbol{\mu}_b$ is the centroid of class $y_b$. Provided that the centroids of different classes are not highly overlapping in most cases, optimizing $\text{sim}_g^{pos}$ over all individual class members is geometrically equivalent (in terms of gradients) to optimizing the similarity to the single class centroid, since the second term is constant with respect to $\Phi(\mathbf{x_b})$. Given that $\nabla_{\Phi(\mathbf{x_b})}$ of the second term vanishes, aggregating pairwise similarities is exactly gradient-equivalent to using a single class centroid (treated as a stop-gradient quantity during optimization). The approximated similarity thus inherits all theoretical properties of the pairwise form, which is a gradient-exact reduction that parallels physical mean-field approximations and conceptually aligns with estab-

lished paradigms like Prototypical Contrastive Learning (Li et al., 2021). More empirical results can be found in Appendix D.2.

Motivated by this rigorous mathematical equivalence and theoretical inspirations, we propose an efficient surrogate approximation using parameterized Gaussian distributions $\mathcal{N}(\boldsymbol{\mu}_m, \boldsymbol{\Sigma}_m)$ for each cluster $m$ to enable efficient boundary embedding shaping with $\mathcal{O}(N)$ complexity. For each evaluated boundary node $v_b$, rather than calculating the computationally heavy pairwise similarities, we use its gravity to the approximated category center $\mu_b$ as the proxy positive similarity measure for the loss function in Thm. 3.1. This precise approximation trains the attention layer $\mathcal{A}^{(B)}$ for structural disentanglement:

$$\text{sim}_g^{pos}(\Phi(\mathbf{x_b}), \mu_b) = -\left[ \max\left(0, \|\Phi(\mathbf{x_b}) - \mu_b\|_2 \right. \right.$$
$$\left. \left. - \min\left\{ \|\Phi(\mathbf{x_i}) - \mu_b\|_2 \mid \mathbf{x_i} \in \mathcal{B} \right\} \right) \right]^2. \quad (9)$$

For the negative similarity measure, we identically use the centers of other categories $\mu_j$ ($j \neq b$) as the negative repelling targets:

$$\text{sim}_g^{neg}(\Phi(\mathbf{x_b}), \mu_j) = -\|\Phi(\mathbf{x_b}) - \mu_j\|_2^2. \quad (10)$$

By utilizing this Gaussian-center-approximated similarity term, the improvements of the attention layer $\mathcal{A}^{(B)}$ will focus securely and efficiently on boundary nodes incorrectly classified by the pre-trained GNN encoder, firmly bridging the theoretical guarantees with operational practice.

### 4.3. Adaptive Structural Disentanglement

Practical disentanglement via contrastive learning processes often encounters instability and feature-level coupling. Unstable updates affect the majority of node embeddings through considerable perturbations of the model parameters. We, instead, utilize an adaptive learning rate to avoid massive perturbations of the attention layer $\mathcal{A}^{(B)}$, which is indicated as feasible by the properties of Section 3.3.

To control the update magnitude of the boundary attention parameters $\boldsymbol{\theta}^{(B)}$, we employ a pre-update mechanism during training. At each iteration, we randomly sample $\beta$ nodes from the boundary set $\mathcal{B}$ to compute the loss gradient $\nabla_{\boldsymbol{\theta}^{(B)}}\mathcal{L}$.

We first perform a virtual pre-update to obtain temporary parameters:

$$\boldsymbol{\theta}'^{(B)} = \boldsymbol{\theta}^{(B)} + \eta \nabla_{\boldsymbol{\theta}^{(B)}}\mathcal{L}, \quad (11)$$

where $\eta$ is the learning rate. This pre-update induces an embedding change across all nodes, measured by:

$$\Delta^{(B)} = \sum_{v_i \in \mathcal{V}} \left\| \mathcal{A}^{(B)}(\Phi(\mathbf{X}); \boldsymbol{\theta}'^{(B)}) - \mathcal{A}^{(B)}(\Phi(\mathbf{X}); \boldsymbol{\theta}^{(B)}) \right\|_2,$$
$$(12)$$

where $\mathcal{A}^{(B)}(\cdot; \boldsymbol{\theta})$ denotes the boundary attention layer parameterized by $\boldsymbol{\theta}$, and $\Phi(\mathbf{X})$ represents the node features.

The actual parameter update is then adaptively scaled:

$$\boldsymbol{\theta}^{(B)} \leftarrow \boldsymbol{\theta}^{(B)} + \frac{\alpha}{\Delta^{(B)}} \cdot \eta \nabla_{\boldsymbol{\theta}^{(B)}}\mathcal{L}, \quad (13)$$

where $\alpha$ is a scaling hyperparameter. The inverse relationship with $\Delta^{(B)}$ ensures that larger embedding perturbations result in smaller parameter updates, preventing excessive changes that could inadvertently push more nodes into boundary regions. We only accept updates where $\Delta^{(B)}$ achieves the minimum change necessary to separate the selected boundary nodes. Notably, the bound $\mathcal{O}(\sqrt{\epsilon}) + \mathcal{O}\left(\frac{1}{\text{margin}(\cdot)}\right)$ derived in Prop. 3.1 has two terms. The inversely scaling updates by $\Delta^{(B)}$ prevent overshooting during optimization, decreasing $\epsilon$; and suppress destabilizing perturbations at boundary nodes, preserving the decision-boundary margin to decrease $\frac{1}{\text{margin}(\cdot)}$. Thus, adaptive scaling is regarded as a theoretically principled operationalization of Prop. 3.1.

The complete computation procedure with complexity analysis is detailed in Algorithm 1 (See Appendix B), producing the improved embeddings $\hat{\mathbf{Z}}^{(B)} = \mathcal{A}^{(B)}(\Phi(\mathbf{X}); \boldsymbol{\theta}^{(B)})$. In addition, the aggregation of GNN tends to blur node distinctions by sharing parameters globally (Wang et al., 2023). We adopt iterative optimization by sequentially separating $\mathcal{A}^{(B)}$ into $\mathcal{A}_1^{(B)} \cdots \mathcal{A}_L^{(B)}$ with each layer trained separately to counteract this effect. It is proved practically useful and provides evidence that parameter separation can prevent feature-level coupling and make $\mathcal{A}^{(B)}$ more effective for structural disentanglement.

BES offers two principal advantages. First, its separating property produces highly distinguishable node embeddings, thereby enhancing the discriminability of nodes, particularly beneficial for identifying boundary nodes. Second, its intrinsic noise-filtering mechanism effectively eliminates irrelevant structural information, which in turn contributes to improved performance in downstream tasks.

## 5. Experiments

Our experiments demonstrate that BES consistently improves classification accuracy across diverse benchmarks. We first show that BES outperforms leading methods on the widely used datasets (Section 5.1), achieving up to 5.0% improvement on WikiCS for node classification. We then analyze BES (Section 5.2) through visualized processing

and embedding changes, detailed ablations, and validate the structural disentanglement from a GNN decoder perspective. Finally, we examine hyperparameter sensitivity and computational efficiency, confirming that BES achieves substantial gains with minimal overhead.

**Datasets and Experiment Setup.** We evaluated different methods on homogeneous (WikiCS (Mernyei & Cangea, 2020), Cora (Pennington et al., 2014), Pubmed (Sen et al., 2008), and CiteSeer (Giles et al., 1998)), heterogeneous (Pei et al., 2020) (Chameleon, Cornell, Texas), heterophilic (Platonov et al., 2023) (Roman-empire), and massive industrial OGB graphs (Hu et al., 2020) (ogbn-arxiv, ogbl-collab). On these datasets, we can comprehensively evaluate the performance of our proposed method from multiple perspectives, which demonstrates not only its advantages in classical benchmarks but also its applicability in more complex scenarios, thereby showcasing the robustness and versatility of the model.

We follow the experimental setup of (Hoang & Lee, 2024) and evaluate the accuracy of different methods on all non-OGB datasets using random 60%/20%/20% train/validation/test splits. The OGB experiment follows the official OGB evaluation protocol. Random seeds are used to ensure the reproducibility of the experiments. The experimental results are averaged over 5 different runs. More details of the datasets, models, and experiment setup are in Appendix C.

**Baselines.** We conduct comparisons between the proposed methods and both classical and state-of-the-art baselines across multiple datasets. Classic methods, such as GCN (Yang et al., 2016), KGNN (Morris et al., 2019), GAT (Velickovic et al., 2018), GATv2 (Brody et al., 2022), GraphSAGE (Hamilton et al., 2017), Cheb (Defferrard et al., 2016), SGC (Wu et al., 2019), and SSGC (Zhu & Koniusz, 2021) are included as baselines. Recent methods, including H2GCN (Zhu et al., 2020), SGNN (Zhang et al., 2024a), MPNN-GCN, MPNN-GAT, MPNN-SAGE (Luo et al., 2024), NodeImport-GCN, NodeImport-GAT, NodeImport-SAGE(Chen et al., 2025a), and contrastive learning methods including GraphECL(Xiao et al., 2024), MotifRGC(Sun et al., 2024), IGCL(Chen et al., 2025b), are compared to show the advantage of our method. Please see Appendix C.4 for more detailed baseline information.

## 5.1. Method Comparison

Table 1 presents node classification and link prediction results on homogeneous datasets. Additional AUC and F1 results are in Appendix D.3, and the results on more challenging heterogeneous, heterophilic, and OGB datasets are presented in Appendix D.4. BES achieves top performance across all benchmarks on both tasks, demonstrating that boundary-focused embedding shaping effectively filters spu-

rious structural noise in both node-level and edge-level representations.

**Node Classification.** Traditional GNNs (GCN, GAT) entangle noise from neighbors and graph propagation within shared parameters, forcing them to process both task-relevant signals and spurious structural noise simultaneously. While SGNN decouples these components, it still struggles with fine-grained boundary discrimination in dense neighborhoods. BES addresses this by employing specialized modules across different embedding shaping stages: initial encoders extract semantic features while boundary-focused contrastive learning selectively refines decision boundaries. This architectural decoupling enables the two components to capture complementary semantic and structural characteristics independently, thereby yielding more expressive representations.

**Link Prediction.** Edge embeddings inherit noise from their constituent node embeddings. By filtering spurious structural correlations at the node level, BES produces cleaner edge representations that better capture genuine connectivity patterns. This explains consistent improvements in link prediction, particularly for edges spanning class boundaries where structural noise is most pronounced.

**Key Advantage.** Unlike methods that uniformly process all nodes, BES explicitly targets boundary regions where structural entanglement causes maximum damage. By systematically disentangling task-relevant topology from spurious correlations through adaptive contrastive learning with controlled parameter updates, BES overcomes fundamental limitations of both traditional GNNs and recent disentangled methods. Results demonstrate that this strategy consistently improves accuracy, validating our theoretical analysis.

## 5.2. Method Analysis

**Boundary Embedding Shaping Process** Figure 3(1) visualizes how BES refines decision boundaries on WikiCS for initial embeddings (embeddings from different graph encoders are shown in Appendix D.5). The attention layers successfully disentangle structurally similar but semantically distinct nodes, pushing previously ambiguous embeddings away from class boundaries. For nodes sharing topological patterns but differing in classes, BES filters spurious structural correlations, allowing the contrastive loss to correct representations that were entangled by the initial GNN encoder. This validates our hypothesis that boundary-focused refinement addresses the core challenge of structural entanglement. The effect of using different shift score calculations, and the comparison of using the exact pairwise objective and its approximation are in Appendix D.1 and Appendix D.2, respectively. More visualization results of BES are in Appendix D.6.

*Table 1.* Node classification (NC) and link prediction (LP) accuracy (↑) for various methods on different datasets. OOM indicates out of memory; - indicates the method does not support the task. The 1st (bold) and 2nd (underlined) results are highlighted.

| Method | Cora | | CiteSeer | | Pubmed | | WikiCS | |
|---|---|---|---|---|---|---|---|---|
| | NC | LP | NC | LP | NC | LP | NC | LP |
| KGNN (Morris et al., 2019) | 87.49 | 85.15 | 76.14 | 87.65 | 89.35 | 85.81 | 28.06 | - |
| GCN (Yang et al., 2016) | 87.74 | 92.30 | 75.74 | 95.30 | 87.36 | 93.70 | 80.48 | 91.89 |
| Cheb (Defferrard et al., 2016) | 87.31 | 90.42 | 74.69 | 89.20 | 89.80 | 87.34 | 82.52 | 90.85 |
| GAT (Velickovic et al., 2018) | 87.18 | 90.39 | 76.24 | 92.35 | 86.81 | 89.07 | 80.88 | 90.87 |
| SGC (Wu et al., 2019) | 88.29 | 91.65 | 77.99 | 94.31 | 87.71 | 86.94 | 80.11 | 92.00 |
| SSGC (Zhu & Koniusz, 2021) | 87.86 | 91.44 | 78.10 | 94.26 | 88.33 | 89.26 | 80.28 | 91.55 |
| GraphSAGE (Hamilton et al., 2017) | 87.74 | 90.67 | 76.79 | 91.59 | 88.86 | 88.30 | 81.30 | 90.15 |
| GATv2 (Brody et al., 2022) | 86.32 | 89.34 | 75.74 | 93.14 | 87.11 | 88.07 | 81.71 | 90.12 |
| H2GCN (Zhu et al., 2020) | 88.69 | 92.27 | 77.74 | 94.97 | 88.77 | 95.73 | 76.27 | 92.04 |
| SGNN (Zhang et al., 2024a) | 87.74 | 93.38 | 77.34 | 96.07 | 83.46 | 94.60 | 74.36 | 91.01 |
| NodeImport-GCN (Chen et al., 2025a) | 82.14 | 87.22 | 67.73 | 85.39 | 84.22 | 85.14 | 78.12 | 89.56 |
| NodeImport-GAT (Chen et al., 2025a) | 81.18 | 86.97 | 66.05 | 82.95 | 84.17 | 84.65 | 80.03 | 89.43 |
| NodeImport-SAGE (Chen et al., 2025a) | 79.96 | 80.07 | 68.36 | 77.72 | 86.84 | 78.39 | 81.26 | 86.41 |
| MotifRGC-GCN (Sun et al., 2024) | 87.73 | 71.87 | 75.19 | 86.56 | 88.76 | 93.50 | OOM | OOM |
| MotifRGC-GAT (Sun et al., 2024) | 87.62 | 88.08 | 75.34 | 87.37 | 87.70 | 97.64 | OOM | OOM |
| MotifRGC-SAGE (Sun et al., 2024) | 87.54 | 88.82 | 75.73 | 91.01 | 88.75 | 97.51 | OOM | OOM |
| MPNN-GCN (Luo et al., 2024) | 85.52 | 90.97 | 76.04 | 93.85 | 89.76 | 86.73 | 84.05 | 89.05 |
| MPNN-GAT (Luo et al., 2024) | 86.14 | 89.43 | 75.44 | 89.96 | 87.29 | 85.64 | 85.48 | 85.87 |
| MPNN-SAGE (Luo et al., 2024) | 86.20 | 87.52 | 75.19 | 89.76 | 89.99 | 79.55 | 85.40 | 84.73 |
| GraphECL (Xiao et al., 2024) | 88.72 | 92.73 | 75.39 | 95.22 | 88.18 | 94.13 | 81.28 | 51.99 |
| IGCL (Chen et al., 2025b) | 88.60 | 92.98 | 77.30 | 95.60 | 87.32 | 96.88 | 82.99 | 89.81 |
| BES (Ours) | **89.46** | **94.45** | **78.40** | **96.27** | **91.03** | **97.89** | **85.52** | **92.27** |

**Entangled vs. Disentangled Embeddings** We compare the classification accuracy of entangled and disentangled embeddings on WikiCS by ablating the attention layers of BES (see Figure 3(2)). The results show that the attention layers of BES can improve the initial entangled node embedding for both node classification and link prediction. Specifically, the disentanglement effect becomes smaller as the number of attention layers increases, which suggests that BES provides most benefits early in the shaping process, after which additional layers yield marginal gains. Full ablation results are provided in Appendix D.7.

**Structural Disentanglement of Final Embeddings** Figure 3(3) visualizes final embeddings from the Cheb GNN decoder. The decoder successfully learns edge weights that filter irrelevant neighbor information, producing well-separated class clusters. This structural disentanglement directly explains the superior accuracy in Section 5.1, confirming that end-to-end boundary shaping yields robust node representations.

**Hyperparameter Sensitivity and Convergence** Figure 3(4) presents sensitivity analysis. Performance is robust across hyperparameters, with optimal settings at $\tau = 1$ and $\delta = 5$. Larger $\beta$ (more boundary nodes) and smaller $\alpha$ (gentler updates) improve accuracy, consistent with our adaptive update mechanism. BES converges under different hyperparameter settings on all datasets, indicating that the adaptive update mechanism is effective. Additional efficiency analysis is provided in Appendix D.8.

## 6. Related Work

**Graph Representation Learning.** Graph Neural Networks (GNNs) such as GCN (Kipf & Welling, 2017), Graph-SAGE (Hamilton et al., 2017), and GAT (Velickovic et al., 2018) have laid the foundation for graph representation learning. However, these models often struggle to filter noisy or irrelevant signals introduced by the complex and diverse topological structures in real-world graphs. GCN suffers from over-smoothing in deeper layers, leading to indistinguishable node representations (Kipf & Welling, 2017). GraphSAGE's performance is highly sensitive to aggregation strategies (Zeng et al., 2020), while GAT introduces significant computational overhead without effectively suppressing topological noise (Velickovic et al., 2018). Recent efforts to improve scalability, such as GraphSAINT (Zeng et al., 2020), or to enhance robustness via contrastive learning (e.g., GraphECL (Xiao et al., 2024), IGCL (Chen et al., 2025b)) offer partial solutions, but they do not explicitly disentangle informative features from noisy graph structure. Transformer-based architectures in graph learning, such as Graphormer (Ying et al., 2021), leverage self-attention to capture long-range dependencies, but they often overlook lo-

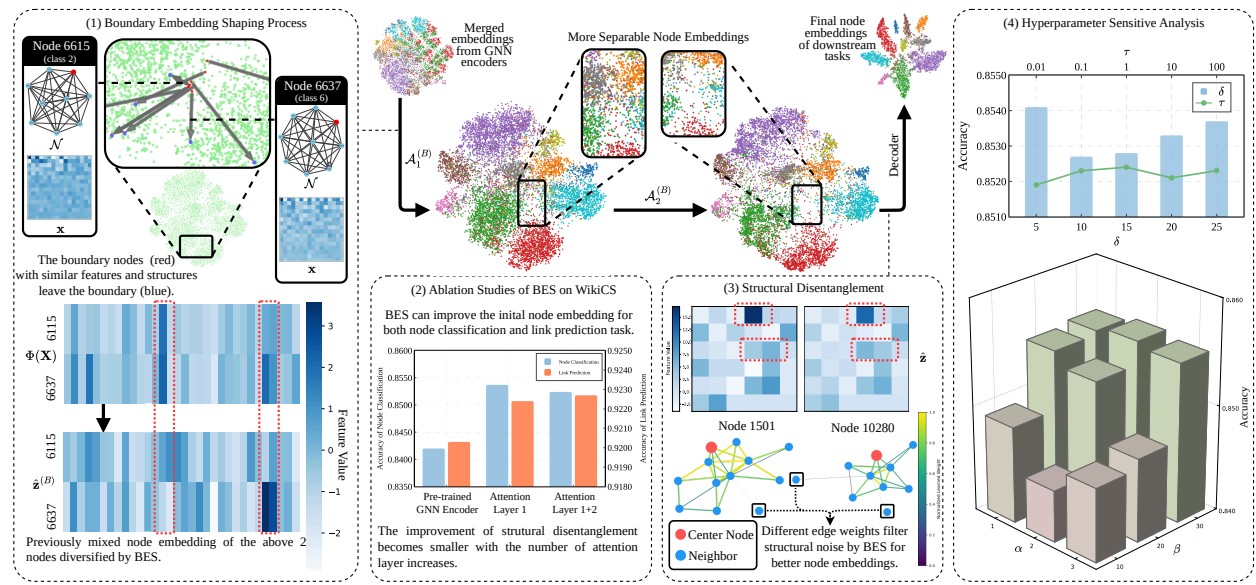

Figure 3. Analysis of boundary embedding shaping.

calized noise and exhibit high computational costs on large graphs.

**Contrastive Representation Learning.** Class-guided contrastive learning has become a cornerstone of graph representation learning, with seminal works such as Supervised Contrastive Learning (SCL) (Khosla et al., 2020), Prototypical Contrastive Learning (PCL) (Li et al., 2021), and GraphECL (Xiao et al., 2024). These methods avoid exhaustive pairwise comparisons by employing a set of cluster prototypes (or centroids) as anchors to approximate global feature distributions, achieving computational efficiency and stable optimization. Motivated by the prototype-based paradigm, BES adopts center-based similarity to efficiently capture structural entanglement and further designs an adaptive pre-update mechanism to selectively filter noisy topological structures.

**Identifiability and Causal Modeling.** Recent work on identifiable latent causal models (Liu et al., 2024; 2026c;a;b; 2025; Bao et al., 2025) provides a theoretical foundation for disentangling latent structures under limited supervision. These studies inspire our BES approach, suggesting that robust latent representations can be achieved by leveraging class-guided signals and sufficient structural variability.

**Disentangled Graph Representation Learning.** Disentangled representation learning on graphs aims to decompose node embeddings into independent factors (feature-level disentanglement) or separate task-relevant subgraphs from noise (structure-level disentanglement) (Ma et al., 2019). Feature-level entanglement is primarily caused by shared model parameters of GNNs, which existing methods

attempt to resolve by introducing multi-channel graph encoders and regularization objectives to encourage factorized latent representations (Ma et al., 2019). However, these approaches still ignore neighbor noise during message passing, leading to cross-node interference that limits their ability to isolate the sources of entanglement. Other works focus on structure-level disentanglement by identifying task-relevant subgraphs or suppressing spurious edges (Miao et al., 2022). Despite their effectiveness, feature encoding and structure learning are often optimized jointly, and most existing approaches lack a principled separation between feature-level and structure-level entanglement, motivating the need for frameworks that explicitly control feature coupling to better study structure-induced interference.

## 7. Conclusion

In this work, we tackled the challenge of graph structural entanglement by revealing boundary nodes as the principal sources of topological ambiguity, where irrelevant topological signals deteriorate representation learning. Building upon this insight, we introduced Boundary Embedding Shaping (BES), a general-purpose plug-in module that leverages adaptive contrastive learning to disentangle predictive structural information from spurious correlations. Theoretical analysis and empirical results confirm that BES effectively refines decision boundaries, leading to significant performance gains on standard benchmarks. Furthermore, extending BES to low-label and self-supervised settings, where class-guided signals are scarce or absent, represents a promising avenue for broadening its applicability to large-scale graphs with limited annotation.

## Impact Statement

This paper presents work whose goal is to filter the noisy topological information as much as possible throughout the learning process to obtain a higher-quality embedding for node classification. The work can have potential benefits in various domains, from social network analysis to molecular structure prediction. We foresee no negative societal consequences.

## Acknowledgements

This work was supported by the Yunnan Fundamental Research Project (202301AU070147).

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

# Appendix

# A. Proofs of Main Results

## A.1. Proof of Thm. 3.1

In this section, we provide a detailed proof of Thm. 3.1. To begin, we restate the theorem for clarity.

**Theorem 3.1** (Isolating Invariant Latent Variables). *Let $(\mathbf{x}, \tilde{\mathbf{x}})$ be a positive pair of observations generated according to the LVM formalized in Section 3.1, and suppose the positive pair construction process follows Asms. 3.1 and 3.2. Let $\mathbf{f} : \mathcal{X} \to (0,1)^{n_{inv}}$ be a sufficiently flexible and smooth function, where $(0,1)^{n_{inv}}$ represents a Cartesian product of open intervals in $n_{inv}$-dimensional space. Then, minimizing the following contrastive loss function with the temperature parameter set to $\tau = 1$ (similar to SimCLR (Chen et al., 2020)) over a sufficient number of sample pairs $(\mathbf{x}, \tilde{\mathbf{x}})$ ensures that $\mathbf{f}$ block-identifies $\mathbf{z}_{inv}$:*

$$\mathcal{L}(\mathbf{z}, \tilde{\mathbf{z}}) = -\log \frac{\exp\left(\text{sim}^{pos}(\mathbf{z}, \tilde{\mathbf{z}})/\tau\right)}{\sum_{\mathbf{z}' \in Z, \mathbf{z}' \neq \mathbf{z}} \exp\left(\text{sim}^{neg}(\mathbf{z}, \mathbf{z}')/\tau\right)}$$

*where $\text{sim}(\cdot, \cdot)$ is a similarity function. Therefore, the learned node embeddings include all and only the information of the invariant latent variables $\mathbf{z}_{inv}$.*

*Proof.* Our proof follows a similar approach to that of (von Kügelgen et al., 2021), adapted to our specific context. The proof consists of four key steps:

- First, we derive the asymptotic form of the contrastive loss defined in Thm. 3.1 and demonstrate that this loss can attain a global minimum of zero.

- Second, we establish that any smooth function achieving this global minimum must satisfy an alignment condition, which ensures that the learned representations of two views in a positive pair are aligned across all sample pairs.

- Third, using a contradiction, we prove that the learned representation cannot contain any component from the variant latent block.

- Finally, we complete the proof by demonstrating that the learned representation includes all information from the invariant latent block.

**Step 1.** As shown in previous works (von Kügelgen et al., 2021; Daunhawer et al., 2023), given infinite samples and when $\tau = 1$, the contrastive loss in Thm. 3.1 can asymptotically approach the following loss:

$$\mathcal{L}_{\text{AlignMaxEnt}}(\mathbf{f}) = \mathbb{E}_{(\mathbf{x}, \tilde{\mathbf{x}}) \sim p_{\mathbf{x}, \tilde{\mathbf{x}}}}\left[\|\mathbf{f}(\mathbf{x}) - \mathbf{f}(\tilde{\mathbf{x}})\|_2\right] - H\left(\mathbf{f}(\mathbf{x})\right), \tag{14}$$

where $H$ denotes the differential entropy.

Furthermore, it is known that the global minimum of this loss is 0, as there exists a smooth function such that

$$\mathbf{f}^* = \mathbf{d} \circ (\mathbf{g}^{-1})_{1:n_{inv}} : \mathcal{X} \to (0,1)^{n_{inv}}. \tag{15}$$

Here, $\mathbf{g}$ denotes the true underlying generative process as defined in Section 3.1; the operator $(\cdot)_{1:n_{inv}}$ extracts the components corresponding to the invariant latent variables; and $\mathbf{d}$ is the Darmois construction (von Kügelgen et al., 2021), where for each $i \in \{1, \ldots, n_{inv}\}$:

$$d_i(\mathbf{z}_{inv}) = \text{CDF}_i(z_{inv,i} \mid \mathbf{z}_{inv,[1:i-1]})$$
$$= \mathbb{P}(Z_{inv,i} \leq z_{inv,i} \mid \mathbf{z}_{inv,[1:i-1]}),$$

with $\text{CDF}_i$ denoting the conditional cumulative distribution function of $z_{inv,i}$ given $\mathbf{z}_{inv,[1:i-1]}$.

To prove this, we simply substitute the candidate function into Equation (14) and obtain

$$\mathcal{L}_{\text{AlignMaxEnt}}(\mathbf{f}^*) = \mathbb{E}_{(\mathbf{x}, \tilde{\mathbf{x}}) \sim p_{\mathbf{x}, \tilde{\mathbf{x}}}} \big[ \| \mathbf{d}(\mathbf{z}_{inv}) - \mathbf{d}(\tilde{\mathbf{z}}_{inv}) \|_2 \big]$$
$$- H\big( \mathbf{d}(\mathbf{z}_{inv}) \big). \tag{16}$$

The substitution is justified by the invertibility of $\mathbf{g}^{-1}$ as formalized in Section 3.1, which ensures that $(\mathbf{g}^{-1})_{1:n_{inv}}$ is well-defined. Furthermore, based on Asm. 3.1, we know that $\tilde{\mathbf{z}}_{inv} = \mathbf{z}_{inv}$ almost surely across all positive pairs, guaranteeing that the first term on the right-hand side of Equation (16) equals 0.

Moreover, by the properties of the Darmois function (von Kügelgen et al., 2021), the mapping $\mathbf{d}$ transforms $\mathbf{z}_{inv}$ into a uniform distribution over $(0, 1)^{n_{inv}}$. Since the uniform distribution is the maximum entropy distribution, which has an entropy of 0, this reduces the second term of Equation (16). Therefore, we have demonstrated that $\mathcal{L}_{\text{AlignMaxEnt}}$ has a global minimum at 0.

**Step 2.** We now prove for any smooth function $\mathbf{f} : \mathcal{X} \to (0, 1)^{n_{inv}}$ that achieves the global minimum of $\mathcal{L}_{\text{AlignMaxEnt}}$, it ensures an alignment condition. First, we define the following smooth mapping:

$$\mathbf{r} := \mathbf{f} \circ \mathbf{g}.$$

Since $\mathcal{L}_{\text{AlignMaxEnt}}$ in Equation (14) has a global minimum at 0 and all of its terms are non-negative. Given $\mathbf{f}$ achieve its minimum, we must have

$$\mathbb{E}_{(\mathbf{x}, \tilde{\mathbf{x}}) \sim p_{\mathbf{x}, \tilde{\mathbf{x}}}} \big[ \| \mathbf{r}(\mathbf{z}) - \mathbf{r}(\tilde{\mathbf{z}}) \|_2 \big] = 0, \tag{17}$$
$$H\big( \mathbf{r}(\mathbf{z}) \big) = 0. \tag{18}$$

From Equation (17), it follows that

$$\mathbf{r}(\mathbf{z}) = \mathbf{r}(\tilde{\mathbf{z}}) \quad \text{almost surely} \quad \forall \mathbf{z} \sim p_{\mathbf{z}}, \ \tilde{\mathbf{z}} \sim p_{\tilde{\mathbf{z}}|\mathbf{z}}. \tag{19}$$

which we term the *invariance condition*.

**Step 3.** Now, we exclude the information of variant latent variables $\mathbf{z}_{var}$ from the learned representations by contradiction.

Suppose, for the sake of contradiction, that there exists a function $\mathbf{r}^c = \mathbf{f}^c \circ \mathbf{g}$ that depends on at least one component of the variant latent variables $\mathbf{z}_{var}$. Formally,

$$\exists i \in \{n_{inv} + 1, \dots, n\}, \ (\mathbf{z}^*_{inv}, \mathbf{z}^*_{var}),$$
$$\text{such that} \quad \frac{\partial \mathbf{r}^c}{\partial z_{var,i}}(\mathbf{z}^*_{inv}, \mathbf{z}^*_{var}) \neq 0.$$

By the $C^1$ continuity of $\mathbf{r}^c$, guaranteed by the smoothness of both $\mathbf{f}^c$ and $\mathbf{g}$, there exists a sufficiently small $\eta > 0$ such that the following inequality holds:

$$\mathbf{r}^c\big( \mathbf{z}^*_{inv}, (\mathbf{z}^*_{var \setminus \{i\}}, z_i^-) \big) \neq \mathbf{r}^c\big( \mathbf{z}^*_{inv}, (\mathbf{z}^*_{var \setminus \{i\}}, z_i^+) \big),$$
$$\forall z_i^- \in (z_i^* - \eta, z_i^*), \ z_i^+ \in (z_i^*, z_i^* + \eta). \tag{20}$$

On the other hand, based on Asm. 3.2, we know that $p_{\tilde{\mathbf{z}}_{var} | \mathbf{z}_{var}}(\cdot \mid \mathbf{z}^*_{var})$ is smooth and strictly positive on $\mathcal{O}(\mathbf{z}^*_{var})$. Thus, there exists a sufficiently small $\eta_1 > 0$ such that

$$p_{\tilde{\mathbf{z}}_{var} | \mathbf{z}_{var}}(\tilde{\mathbf{z}}_{var} \mid \mathbf{z}^*_{var}) > 0,$$
$$\forall \tilde{\mathbf{z}}_{var} \in \{\mathbf{z}^*_{var \setminus \{i\}}\} \times (z^*_{var,i} - \eta_1, z^*_{var,i} + \eta_1),$$

with $\{\mathbf{z}^*_{var \setminus \{i\}}\} \times (z^*_{var,i} - \eta_1, z^*_{var,i} + \eta_1) \subseteq \mathcal{O}(\mathbf{z}^*_{var})$. Furthermore, since $p_{\mathbf{z}}$ is positive almost everywhere, so is $p_{\mathbf{z}_{inv}}$. Therefore, we can find two distinct realizations of $\tilde{\mathbf{z}}_{var}$ as the positive counterpart of $(\mathbf{z}^*_{inv}, \mathbf{z}^*_{var})$:

$$\big( \mathbf{z}^*_{inv}, (\mathbf{z}^*_{var \setminus \{i\}}, z_i') \big), \quad \big( \mathbf{z}^*_{inv}, (\mathbf{z}^*_{var \setminus \{i\}}, z_i'') \big),$$

where

$$z_i' \in (z_i^* - \eta_2, z_i^*), z_i'' \in (z_i^*, z_i^* + \eta_2)$$

with

$$\eta_2 = \min(\eta, \eta_1).$$

Based on the invariance condition established in Equation (19), we have the following equalities:

$$\mathbf{r}^c(\mathbf{z}_{inv}^*, \mathbf{z}_{var}^*) = \mathbf{r}^c\big(\mathbf{z}_{inv}^*, (\mathbf{z}_{var\setminus\{i\}}^*, z_i')\big)$$
$$= \mathbf{r}^c\big(\mathbf{z}_{inv}^*, (\mathbf{z}_{var\setminus\{i\}}^*, z_i'')\big).$$

However, this contradicts the inequality established in Equation (20). This contradiction indicates that such an $\mathbf{r}^c$ cannot exist. Therefore, any $\mathbf{r}$ minimizing the loss in Equation (14) must be independent of the variant latent variables, formally:

$$\mathbf{r}(\mathbf{z}) = \mathbf{r}(\mathbf{z}_{inv}) \qquad \text{almost surely} \qquad \forall \mathbf{z} \sim p_{\mathbf{z}}. \tag{21}$$

This result implies that the learned representations contain only information about the invariant latent variables $\mathbf{z}_{inv}$ for any input samples.

**Step 4.** We define the learned representations as $\hat{\mathbf{z}}$. Based on Equation (21), we have

$$\hat{\mathbf{z}} = \mathbf{r}(\mathbf{z}_{inv}) \qquad \text{with} \qquad \hat{\mathbf{z}} \in (0,1)^{n_{inv}}.$$

Furthermore, from Equation (18), we know that the learned representations are uniformly distributed over $(0,1)^{n_{inv}}$. By directly applying Lemma B.2 in (Cai et al., 2025), we conclude that the learned representations contain *all* and *only* the information about the invariant latent variables $\mathbf{z}_{inv}$ almost surely. In other words, $\mathbf{z}_{inv}$ is block-identified by $\mathbf{f}$ in the sense of Defn. 3.1 when minimizing the loss.

This completes the proof of Thm. 3.1. $\qquad\qquad\qquad\qquad\qquad\qquad\qquad\qquad\qquad\qquad\qquad\qquad\quad$ $\square$

## A.2. Proof of Cor. 3.1

In this section, we provide a detailed proof of Cor. 3.1. To begin, we restate the corollary for clarity.

**Corollary 3.1** (Structure-Level Disentanglement). *Minimizing the contrastive loss under Asms. 3.1 and 3.2 maximizes the mutual information between the node embeddings $\hat{\mathbf{z}}$ and the invariant structure $\mathbf{z}_{inv}$, while minimizing information about the variant topology $\mathbf{z}_{var}$:*

$$\max_{\mathbf{f}} I(\hat{\mathbf{z}}; \mathbf{z}_{inv}) \quad s.t. \quad I(\hat{\mathbf{z}}; \mathbf{z}_{var}) \to 0.$$

*This confirms that the optimal node embeddings achieve structure-level disentanglement, effectively filtering structural noise.*

*Proof.* We prove that minimizing the contrastive loss under Assumptions 3.1 and 3.2 leads to structure-level disentanglement by establishing both the maximization of $I(\hat{\mathbf{z}}; \mathbf{z}_{inv})$ and the minimization of $I(\hat{\mathbf{z}}; \mathbf{z}_{var})$.

**Part 1: Maximizing $I(\hat{\mathbf{z}}; \mathbf{z}_{inv})$.** From Theorem 3.1, we know that minimizing the contrastive loss ensures block-identifiability of $\mathbf{z}_{inv}$, meaning there exists an invertible function $\mathbf{r}$ such that $\hat{\mathbf{z}} = \mathbf{r}(\mathbf{z}_{inv})$.

Since $\mathbf{r}$ is invertible, we have:

$$I(\hat{\mathbf{z}}; \mathbf{z}_{inv}) = H(\mathbf{z}_{inv}) - H(\mathbf{z}_{inv}|\hat{\mathbf{z}}). \tag{22}$$

The invertibility of $\mathbf{r}$ implies that $\mathbf{z}_{inv}$ is fully determined by $\hat{\mathbf{z}}$, hence $H(\mathbf{z}_{inv}|\hat{\mathbf{z}}) = 0$. Therefore:

$$I(\hat{\mathbf{z}}; \mathbf{z}_{inv}) = H(\mathbf{z}_{inv}), \tag{23}$$

which is the maximum possible mutual information, achieved when $\hat{\mathbf{z}}$ contains all information about $\mathbf{z}_{inv}$.

**Part 2: Minimizing $I(\hat{\mathbf{z}}; \mathbf{z}_{var})$.** By the block-identifiability result in Definition 3.1, the learned representation $\hat{\mathbf{z}}$ contains all and only the information about $\mathbf{z}_{inv}$. This means that $\hat{\mathbf{z}}$ is conditionally independent of $\mathbf{z}_{var}$ given $\mathbf{z}_{inv}$:

$$\hat{\mathbf{z}} \perp \mathbf{z}_{var} \mid \mathbf{z}_{inv}. \tag{24}$$

Using the chain rule for mutual information:

$$I(\hat{\mathbf{z}}; \mathbf{z}_{var}) = I(\hat{\mathbf{z}}; \mathbf{z}_{var}, \mathbf{z}_{inv}) - I(\hat{\mathbf{z}}; \mathbf{z}_{inv}). \tag{25}$$

Since $\hat{\mathbf{z}} = \mathbf{r}(\mathbf{z}_{inv})$ is a deterministic function of $\mathbf{z}_{inv}$ only, we have:

$$I(\hat{\mathbf{z}}; \mathbf{z}_{var}, \mathbf{z}_{inv}) = I(\hat{\mathbf{z}}; \mathbf{z}_{inv}) = H(\mathbf{z}_{inv}). \tag{26}$$

Therefore:

$$I(\hat{\mathbf{z}}; \mathbf{z}_{var}) = H(\mathbf{z}_{inv}) - H(\mathbf{z}_{inv}) = 0. \tag{27}$$

This confirms that the optimal embeddings achieve structure-level disentanglement by maximizing information about invariant factors while eliminating information about variant topology. □

### A.3. Proof of Prop. 3.1

In this section, we provide a detailed proof of Prop. 3.1. To begin, we restate the corollary for clarity.

**Proposition 3.1** (Disentanglement Error Bound). *Let $\epsilon = \mathbb{E}[\mathcal{L}]$ be the expected loss. The mean squared error between the learned representation $\hat{\mathbf{z}}$ and the true invariant factors (up to linear transformation A) is bounded by:*

$$\inf_{A} \mathbb{E}[\|\hat{\mathbf{z}} - A\mathbf{z}_{inv}\|^2] \leq \mathcal{O}(\sqrt{\epsilon}) + \mathcal{O}\left(\frac{1}{margin(\partial\mathcal{Z}_{inv})}\right),$$

*where $margin(\partial\mathcal{Z}_{inv})$ denotes the separation margin at the class boundaries in the embedding space.*

*Proof.* We derive an upper bound on the disentanglement error by analyzing the optimization landscape of the contrastive loss and its relationship to the geometric properties of the embedding space.

**Step 1: Relate the loss to representation error.** Let $\mathbf{z}^* = A\mathbf{z}_{inv}$ denote the optimal linear transformation of the true invariant factors. The contrastive loss can be decomposed as:

$$\mathcal{L} = \mathbb{E}_{(\mathbf{x},\tilde{\mathbf{x}})}\left[-\log\frac{\exp\left(\mathrm{sim}(\hat{\mathbf{z}}, \tilde{\hat{\mathbf{z}}})/\tau\right)}{\sum_{\mathbf{z}' \neq \hat{\mathbf{z}}} \exp(\mathrm{sim}(\hat{\mathbf{z}}, \mathbf{z}')/\tau)}\right]. \tag{28}$$

For positive pairs satisfying Assumption 3.1, we have $\mathbf{z}_{inv} = \tilde{\mathbf{z}}_{inv}$, which implies $\mathbf{z}^* = \tilde{\mathbf{z}}^*$. The similarity between learned representations can be expressed as:

$$\mathrm{sim}(\hat{\mathbf{z}}, \tilde{\hat{\mathbf{z}}}) = \mathrm{sim}(\mathbf{z}^*, \tilde{\mathbf{z}}^*) - \|\hat{\mathbf{z}} - \mathbf{z}^*\|^2 - \|\tilde{\hat{\mathbf{z}}} - \tilde{\mathbf{z}}^*\|^2 + o(\|\hat{\mathbf{z}} - \mathbf{z}^*\|^2), \tag{29}$$

where we use the fact that for normalized vectors, $\langle \mathbf{u}, \mathbf{v}\rangle = 1 - \frac{1}{2}\|\mathbf{u} - \mathbf{v}\|^2 + o(\|\mathbf{u} - \mathbf{v}\|^2)$.

**Step 2: Upper bound via first-order optimality.** At the optimum, the gradient of the expected loss with respect to $\hat{\mathbf{z}}$ vanishes. By Taylor expansion around the optimal $\mathbf{z}^*$:

$$\mathbb{E}[\mathcal{L}] \geq \mathbb{E}[\mathcal{L}]|_{\hat{\mathbf{z}}=\mathbf{z}^*} + \frac{\lambda_{\min}}{2}\mathbb{E}[\|\hat{\mathbf{z}} - \mathbf{z}^*\|^2], \tag{30}$$

where $\lambda_{\min}$ is the minimum eigenvalue of the Hessian of the loss function.

Rearranging:

$$\mathbb{E}[\|\hat{\mathbf{z}} - \mathbf{z}^*\|^2] \leq \frac{2}{\lambda_{\min}}\left(\mathbb{E}[\mathcal{L}] - \mathbb{E}[\mathcal{L}]|_{\mathbf{z}^*}\right) \leq \frac{2\epsilon}{\lambda_{\min}}. \tag{31}$$

**Step 3: Connect curvature to margin.** The minimum eigenvalue $\lambda_{\min}$ of the loss Hessian is related to the geometry of the class boundaries. Specifically, for points near the decision boundary $\partial \mathcal{Z}_{inv}$, the curvature of the loss surface is inversely proportional to the margin:

$$\lambda_{\min} \geq c \cdot \mathrm{margin}(\partial \mathcal{Z}_{inv}), \tag{32}$$

for some constant $c > 0$ that depends on the smoothness of the embedding function $\mathbf{f}$ and the data distribution.

This relationship arises because when the margin is small, small perturbations in the embedding can lead to misclassification, requiring the loss to be nearly flat (small curvature) in directions perpendicular to the boundary. Conversely, a large margin allows for steeper loss gradients, corresponding to larger eigenvalues.

**Step 4: Combine bounds.** Substituting the margin-curvature relationship:

$$\mathbb{E}[\|\hat{\mathbf{z}} - \mathbf{z}^*\|^2] \leq \frac{2\epsilon}{c \cdot \mathrm{margin}(\partial \mathcal{Z}_{inv})}. \tag{33}$$

Taking the square root and applying Jensen's inequality:

$$\inf_A \mathbb{E}[\|\hat{\mathbf{z}} - A\mathbf{z}_{inv}\|^2] \leq \mathcal{O}(\sqrt{\epsilon}) + \mathcal{O}\left(\frac{1}{\mathrm{margin}(\partial \mathcal{Z}_{inv})}\right), \tag{34}$$

where the $\mathcal{O}(\sqrt{\epsilon})$ term captures the optimization error and the margin-dependent term reflects the geometric complexity of the class boundary.

This completes the proof, showing that the disentanglement error is bounded by both the optimization quality (through $\epsilon$) and the geometric separability (through the margin). $\qquad\square$

## B. Algorithm of BES and Complexity Analysis

---

**Algorithm 1** Boundary Embedding Shaping

---

1: **Input:** Node embeddings $\Phi(\mathbf{X})$, Number of layers $L$, Iterations $Q$, Batch size $\beta$, Learning rate $\eta$, Scaling factor $\alpha$
2: **Output:** Refined node embeddings $\hat{\mathbf{Z}}^{(B)}$
3: Initialize $\hat{\mathbf{Z}}^{(B)} \leftarrow \Phi(\mathbf{X})$
4: **for** $l = 1$ to $L$ **do**
5:     Estimate cluster parameters $\boldsymbol{\mu}_m, \boldsymbol{\Sigma}_m$ using $\hat{\mathbf{Z}}$
6:     Identify boundary region $\mathcal{B}$ via Equation (5)
7:     Compute shift scores $S(v)$ for $v \in \mathcal{B}$ via Equation (4)
8:     Select boundary nodes $\mathcal{B} = \{v \in \mathcal{B} \mid S(v) > 0.5\}$
9:     Initialize attention parameters $\boldsymbol{\theta}_l^{(B)}$
10:     **for** $t = 1$ to $Q$ **do**
11:         Sample batch $\mathcal{V}_{batch}$ of size $\beta$ from $\mathcal{B}$
12:         Compute gradient $\nabla_{\boldsymbol{\theta}_l^{(B)}} \mathcal{L}$ using $\mathcal{V}_{batch}$
13:         Virtual update: $\boldsymbol{\theta}_l'^{(B)} \leftarrow \boldsymbol{\theta}_l^{(B)} + \eta \nabla_{\boldsymbol{\theta}_l^{(B)}} \mathcal{L}$
14:         Compute perturbation $\Delta^{(B)}$
15:         Adaptive update: $\boldsymbol{\theta}_l^{(B)} \leftarrow \boldsymbol{\theta}_l^{(B)} + \frac{\alpha}{\Delta^{(B)}} \eta \nabla_{\boldsymbol{\theta}_l^{(B)}} \mathcal{L}$
16:     **end for**
17:     Update embeddings: $\hat{\mathbf{Z}}^{(B)} \leftarrow \mathcal{A}_l^{(B)}(\hat{\mathbf{Z}}^{(B)})$
18: **end for**
19: **return** $\hat{\mathbf{Z}}^{(B)}$

---

In this section, we provide a detailed analysis of the time and space complexity of Boundary Embedding Shaping (BES), as described in Algorithm 1. Let $N$ be the number of nodes, $D$ be the embedding dimension, $C$ be the number of classes, $K$ be the average node degree (or neighborhood size), and $Q$ be the number of iterations per layer $L$.

**Time Complexity.** The BES algorithm proceeds layer-by-layer. Within each layer:

1. *Gaussian Parameter Estimation*: Computing the class centers and covariance matrices takes $\mathcal{O}(ND^2 + CD^3)$ time in the worst case (using full covariance inference).

2. *Boundary Identification*: Evaluating Equation (5) involves projecting embeddings onto the covariance-scaled center differences, taking $\mathcal{O}(NCD)$ time.

3. *Shift Score Computation*: Computing shift score $S(v)$ via k-nearest neighbors in the embedding space takes $\mathcal{O}(ND \log N)$ time using tree-based spatial indexing.

4. *Attention Training and Adaptive Update*: For each of the $Q$ iterations, we sample a batch to compute the loss gradient, taking $\mathcal{O}(\beta KD^2)$. However, the most computationally expensive operation is evaluating the global perturbation $\Delta^{(B)}$, which requires performing the attention layer update $\mathcal{A}^{(B)}$ over *all* nodes to calculate the total variation. This takes $\mathcal{O}(NKD^2)$ time per iteration. Thus, the inner loop bounds the layer-wise time complexity at $\mathcal{O}(QNKD^2)$.

Consequently, the overall time complexity of BES is dominated by the iterative adaptive update step, leading to $\mathcal{O}(L \cdot QNKD^2)$. In practical implementation, the computation of the global perturbation $\Delta^{(B)}$ can be efficiently accelerated through GPU-parallelized Sparse Dense Matrix Multiplication (SpMM) and distributed graph partitioning, enabling scalable processing on large-scale graphs.

**Space Complexity.** BES explicitly stores the node embeddings $\mathcal{O}(ND)$, the sparse graph structure $\mathcal{O}(NK)$, the parameters for the Gaussian distributions $\mathcal{O}(CD^2)$, and the trainable parameters for the attention layer $\mathcal{O}(D^2)$. Thus, the overall space complexity is $\mathcal{O}(ND + NK + CD^2)$, which scales linearly with the graph size, demonstrating memory efficiency comparable to standard message-passing networks.

## C. Additional Experimental Details

### C.1. Model Details

We use pre-trained GNN-based encoders $\Phi$ to obtain the initial graph data features $\Phi(\mathbf{X})$. Existing GNN-based encoders use different aggregation $(T)$ and update $(U)$ functions, and the selection of $T_1, \ldots, T_K$ and $U_1, \ldots, U_K$ is ideal for obtaining comprehensive embeddings $\Phi_i$ of node $v_i$ from different views.

Each GNN encoder $\Phi_k$ with $L$ layers encodes the node representation through the message-passing process at each layer, which involves neighbor message aggregation and node message update. The neighbor message aggregation centered at each node $v_i \in \mathcal{V}$, where an aggregation function $T_k$ aggregates information from $v_i$'s neighbors $\mathcal{N}(v_i)$ to obtain the aggregated message $\mathbf{m}_{ik}^{(l)}$. Neighbor nodes $\mathcal{N}(v_i)$ of $v_i$ are:

$$\mathcal{N}(v_i) = \{v_j\}_{e_{ji} \in \mathcal{E}}. \tag{35}$$

Thus, the aggregated message $\mathbf{m}_{ik}^{(l)}$ can be obtained by the aggregation function $T_k$:

$$\mathbf{m}_{ik}^{(l)} = T_k^{(l)} \left( \{\Phi_{jk}^{(l)}(\mathbf{x}_j) : v_j \in \mathcal{N}(v_i)\} \right), \tag{36}$$

where $\Phi_{jk}^{(l)}(\mathbf{x}_j) \in \mathbb{R}^D$ is the embedding of neighbor node $v_j$ at layer $l$ of $k$-th GNN encoder, and its initial value $\Phi_{jk}^{(0)}(\mathbf{x}_j) = \mathbf{x}_j \in \mathbf{X}$. After that, the embedding $\Phi_{ik}^{(l+1)}(\mathbf{x}_i)$ of node $v_i$ is updated by the update function $U_k$ based on the aggregated message $\mathbf{m}_{ik}^{(l)}$:

$$\Phi_{ik}^{(l+1)}(\mathbf{x}_i) = U_k^{(l)} \left( \Phi_i^{(l)}(\mathbf{x}_i), \mathbf{m}_{ik}^{(l)} \right). \tag{37}$$

So we obtain the embedding $\Phi_{jk}(\mathbf{x}) = \Phi_{jk}^{(L)}(\mathbf{x})$ for node $v_i$ by encoder $k$.

By the combination of feature extraction using different GNN-based encoders, we obtain the comprehensive node embeddings from different views, denoted as $\Phi(\mathbf{X})$:

$$\Phi(\mathbf{X}) = \text{concat} \left( \Phi_1(\mathbf{X}), \ldots, \Phi_K(\mathbf{X}) \right), \tag{38}$$

which supports the shaping of node embeddings.

*Table 2.* Dataset statistics.

| Dataset | Nodes | Edges | Feature Dim. | Node Classes |
|---|---|---|---|---|
| WikiCS (Mernyei & Cangea, 2020) | 11,701 | 216,123 | 300 | 10 |
| Cora (McCallum et al., 2000) | 2,708 | 5,429 | 1,433 | 7 |
| Pubmed (Sen et al., 2008) | 19,717 | 44,338 | 500 | 3 |
| CiteSeer (Giles et al., 1998) | 3,327 | 4,732 | 3,703 | 6 |
| Chameleon (Pei et al., 2020) | 2,277 | 36,101 | 2,325 | 5 |
| Cornell (Pei et al., 2020) | 183 | 298 | 1,703 | 5 |
| Texas (Pei et al., 2020) | 183 | 309 | 1,703 | 5 |
| Roman-empire (Platonov et al., 2023) | 22,662 | 32,927 | 300 | 18 |
| ogbn-arxiv (Hu et al., 2020) | 169,343 | 1,166,243 | 128 | 40 |
| ogbl-collab (Hu et al., 2020) | 235,868 | 1,285,465 | 128 | – |

Next, based on the initial node embeddings, we use two attention layers $\mathcal{A}_1^{(B)}$ and $\mathcal{A}_2^{(B)}$, which are trained with gravity loss separately, with residual connections to learn the disentangled features of boundary nodes. The attention layer is parameterized by $\boldsymbol{\theta}$, which computes the relationship between each position from different embeddings $\Phi(\mathbf{x}_i)$ for node $v_i$, and generates refined embeddings based on these relationships.

Let the input of $\mathcal{A}_1^{(B)}$ be $\Phi(\mathbf{X}) \in \mathbb{R}^{N \times D}$, and the input of $\mathcal{A}_2^{(B)}$ is the output of $\mathcal{A}_1^{(B)}$. For each attention head $a$ of each attention layer $\mathcal{A}$,

$$\mathcal{A}_a(\Phi(\mathbf{X})) = \mathrm{softmax}\left(\frac{Q_a K_a^\top}{\sqrt{D_a}}\right) V_a, \tag{39}$$

where $Q_a = \Phi(\mathbf{X}) \cdot W_{Q_a}$, $K_a = \Phi(\mathbf{X}) \cdot W_{K_a}$, $V_a = \Phi(\mathbf{X}) \cdot W_{V_a}$ are the query, key, and value functions of the attention layer, and $D_a$ is the dimension of the embeddings. The outputs from all heads are concatenated to form a vector with a dimension identical to $\Phi(\mathbf{X})$, followed by a linear transformation and residual connection to produce the refined node embeddings for the boundary embedding shaping:

$$\begin{aligned}
\hat{\mathbf{z}}^{(B)} &= \mathcal{A}^{(B)}\big(\Phi(X)\big) \\
&= \mathrm{concat}\Big(\mathcal{A}_1^{(B)}(\Phi(\mathbf{X})), \ \ldots, \ \mathcal{A}_a^{(B)}(\Phi(\mathbf{X}))\Big) W_{out} \\
&\quad + \Phi(\mathbf{X}).
\end{aligned} \tag{40}$$

The Chebyshev convolution (Defferrard et al., 2016) is adopted as the decoder $\Psi$ in our experiments due to its effectiveness in capturing local graph topology. This decoder is applied to the refined embeddings produced by our attention layers for node classification and link prediction.

## C.2. Datasets

We use the following datasets, shown in Table 2, to evaluate different methods comprehensively.

- WikiCS (Mernyei & Cangea, 2020). The WikiCS dataset, derived from Wikipedia categories, is designed for benchmarking Graph Neural Networks. It comprises 10 classes related to computer science branches, with high connectivity. Node features are 300-dimensional vectors, calculated as averages of pre-trained GloVe word embeddings (Pennington et al., 2014) from the article.

- Cora (McCallum et al., 2000). The Cora dataset includes 2,708 publications across seven classes, with 5,429 citation links. Each publication is represented by a 0/1 word vector based on a dictionary of 1,433 unique words.

- CiteSeer(Giles et al., 1998). The CiteSeer dataset contains 3,312 publications in six classes, with 4,732 citation links. Each publication is described by a 0/1 word vector from a 3,703-word dictionary.

*Table 3.* Hyperparameter settings for baselines

| Model | lr | Epoch | Optimizer | Dropout | Weight Decay | Implementation |
|---|---|---|---|---|---|---|
| KGNN | 0.01 | 200 | adam | 0.5 | 0.0005 | torch_geometric |
| GCN | 0.01 | 200 | adam | 0.5 | 0.0005 | torch_geometric |
| Cheb | 0.01 | 200 | adam | 0.5 | 0.0005 | torch_geometric |
| GAT | 0.01 | 200 | adam | 0.5 | 0.0005 | torch_geometric |
| SGC | 0.01 | 200 | adam | 0.5 | 0.0005 | torch_geometric |
| SSGC | 0.01 | 200 | adam | 0.5 | 0.0005 | torch_geometric |
| GraphSAGE | 0.01 | 200 | adam | 0.5 | 0.0005 | torch_geometric |
| GATv2 | 0.01 | 200 | adam | 0.5 | 0.0005 | torch_geometric |
| H2GCN | 0.01 | 500 | adam | 0.5 | 0.0005 | official GitHub repo. |
| SGNN | 0.001 | | adam | | | official GitHub repo. |
| NodeImport-GCN | 0.01 | 500 | adam | | 0.0 | official GitHub repo. |
| NodeImport-GAT | 0.01 | 500 | adam | | 0.0 | official GitHub repo. |
| NodeImport-SAGE | 0.01 | 500 | adam | | 0.0 | official GitHub repo. |
| MotifRGC-GCN | 0.01 | 200 | adam | | 0.0 | official GitHub repo. |
| MotifRGC-GAT | 0.01 | 200 | adam | | 0.0 | official GitHub repo. |
| MotifRGC-SAGE | 0.01 | 200 | adam | | 0.0 | official GitHub repo. |
| IGCL | 0.005 | 100 | adam | | 0.000001 | official GitHub repo. |
| OURS | 0.01 | 200 | adam | 0.5 | 0.0005 | |

*Table 4.* Hyperparameter settings for MPNN variants across different datasets

| Hyperparameter | GNN | WikiCS | Cora | CiteSeer | Pubmed | Chameleon | Cornell | Texas | Roman-empire | ogbn-arxiv |
|---|---|---|---|---|---|---|---|---|---|---|
| Hidden Channels | GCN | 256 | 512 | 512 | 256 | 512 | 256 | 256 | 256 | 256 |
| | SAGE | 256 | 256 | 512 | 512 | 512 | 512 | 512 | 512 | 512 |
| | GAT | 512 | 512 | 256 | 512 | 512 | 512 | 512 | 512 | 512 |
| Learning Rate | GCN | 0.001 | 0.001 | 0.001 | 0.005 | 0.005 | 0.005 | 0.005 | 0.005 | 0.005 |
| | SAGE | 0.001 | 0.001 | 0.001 | 0.005 | 0.005 | 0.005 | 0.005 | 0.005 | 0.005 |
| | GAT | 0.001 | 0.001 | 0.001 | 0.01 | 0.001 | 0.001 | 0.001 | 0.001 | 0.001 |
| Layers | GCN | 3 | 3 | 2 | 2 | 3 | 3 | 3 | 3 | 3 |
| | SAGE | 2 | 3 | 3 | 4 | 4 | 4 | 4 | 4 | 4 |
| | GAT | 2 | 3 | 3 | 2 | 3 | 3 | 3 | 3 | 3 |
| Weight Decay | GCN | 0.0 | 5e-4 | 0.01 | 5e-4 | 5e-4 | 5e-4 | 5e-4 | 5e-4 | 5e-4 |
| | SAGE | 0.0 | 5e-4 | 0.01 | 5e-4 | 5e-4 | 5e-4 | 5e-4 | 5e-4 | 5e-4 |
| | GAT | 0.0 | 5e-4 | 0.01 | 5e-4 | 5e-4 | 5e-4 | 5e-4 | 5e-4 | 5e-4 |

- PubMed(Sen et al., 2008) . The PubMed dataset contains 19,717 diabetes-related publications, divided into three classes. It features a citation network with 44,338 links. Each publication is represented by a TF-IDF weighted word vector from a 500-word dictionary.

- Chameleon (Pei et al., 2020). The Chameleon dataset is a Wikipedia page-page network where nodes represent articles and edges reflect mutual hyperlinks. It comprises 2,277 nodes, 36,101 edges, and 5 node classes. Nodes are classified by average monthly traffic, and node features indicate the presence of informative nouns in the article content.

- Cornell (Pei et al., 2020). The Cornell dataset is part of the WebKB collection, consisting of web pages from Cornell University. It includes 183 nodes, 298 edges, and 5 node classes (student, project, course, staff, and faculty). Node features are bag-of-words vectors of the web page content.

- Texas (Pei et al., 2020). The Texas dataset is part of the WebKB collection, consisting of web pages from the University of Texas at Austin. It contains 183 nodes, 309 edges, and 5 node classes (student, project, course, staff, and faculty). Node features are bag-of-words vectors of the web page content.

- Roman-empire (Platonov et al., 2023). The Roman-empire dataset is constructed from the English Wikipedia article "Roman Empire," where each node represents a word in the article and edges encode sequential order or syntactic

*Table 5.* Hyperparameter settings for GraphECL across different datasets

| Dataset | Epochs | lr1 | lr2 | wd1 | wd2 | Hidden Dim | Temp |
|---------|--------|------|------|------|------|------------|------|
| WikiCS | 1000 | 5e-4 | 1e-2 | 1e-6 | 1e-5 | 1024 | 0.5 |
| Cora | 100 | 5e-4 | 1e-2 | 1e-6 | 1e-5 | 1024 | 0.5 |
| CiteSeer | 100 | 1e-3 | 1e-2 | 1e-5 | 1e-2 | 2048 | 0.5 |
| Pubmed | 100 | 5e-4 | 1e-2 | 0 | 1e-4 | 1024 | 0.5 |

*Table 6.* Accuracy (↑) using graph topology vs embedding space neighbors. The 1st (bold) results are highlighted.

| Settings | Epochs | Cora | Citeseer | Pubmed | WikiCS |
|----------|--------|------|----------|--------|--------|
| **Node Classification** | | | | | |
| Graph Topology | 200 | 89.40±0.20 | 78.18±0.09 | **91.09±0.06** | 85.28±0.04 |
| Embedding Space | 200 | **89.40±0.17** | **78.20±0.06** | 91.03±0.03 | **85.30±0.15** |
| **Link Prediction** | | | | | |
| Graph Topology | 200 | **94.92±0.01** | 95.65±0.09 | 97.76±0.04 | 92.24±0.08 |
| Embedding Space | 200 | 94.92±0.03 | **95.66±0.06** | **97.80±0.02** | **92.24±0.04** |

dependencies. It includes 22,662 nodes, 32,927 edges, and 18 node classes corresponding to syntactic roles. Node features are 300-dimensional pre-trained word embeddings.

- ogbn-arxiv (Hu et al., 2020). The ogbn-arxiv dataset is a directed citation network of CS papers indexed on arXiv. It contains 169,343 nodes, 1,166,243 edges, and 40 node classes corresponding to CS subject areas. Node features are 128-dimensional embeddings obtained by averaging skip-gram word embeddings of each paper's title and abstract.

- ogbl-collab (Hu et al., 2020). The ogbl-collab dataset is an undirected author collaboration network, where nodes represent authors and edges represent co-authorship relations. It contains 235,868 nodes and 1,285,465 edges. Node features are 128-dimensional embeddings obtained by averaging the word embeddings of each author's published papers. This dataset is used for link prediction.

### C.3. Experimental Environment

All experiments on different datasets were conducted on a server equipped with NVIDIA A40 GPUs (48 GB memory per GPU), a 64-core Intel Xeon Silver 4216 CPU, and 256 GB of system memory. The software implementation is based on Python 3.9.18, PyTorch 2.0.0, CUDA 11.7, and PyTorch Geometric 2.4.0.

### C.4. Hyperparameter Settings

Table 3 provides an overview of the hyperparameter settings for various Graph Neural Network (GNN) models in our experiments. We also separately give the detailed hyperparameter settings for MPNN-GNN and GraphECL across different datasets in Table 4 and Table 5, respectively. These settings include learning rate, number of epochs of training, optimizers used, dropout rate, weight decay, and the types of implementation.

## D. Additional Experimental Results

### D.1. Shift Score via Graph Topology vs. Embedding Space Neighbors

How shift score calculations affect node classification and link prediction performance is shown in Table 6. From the results, we can see that selecting neighbors in the embedding space rather than using graph topology yields slightly better performance. Notably, although the shift score is computed in the embedding space, its objective remains firmly anchored in graph topology, as boundary nodes' local neighborhoods are heavily influenced by heterophilic topological structures, showing similar topological shift scores across different graphs. Moreover, we observe that nodes with the noisiest local topology are not always located precisely on the decision boundary. Updating such nodes does not necessarily help the

*Table 7.* Accuracy (↑) using pairwise objective vs center-based approximation. The 1st (bold) results are highlighted to indicate their rankings.

| Settings | epochs | Cora | Citeseer | Pubmed | WikiCS |
|---|---|---|---|---|---|
| **Node Classification** | | | | | |
| Pairwise Objective | 250 | **89.42±0.21** | **78.38±0.19** | 90.99±0.07 | 85.22±0.12 |
| Center-based Approximation | 200 | 89.40±0.17 | 78.20±0.06 | **91.03±0.03** | **85.30±0.15** |
| **Link Prediction** | | | | | |
| Pairwise Objective | 250 | 94.89±0.11 | 95.59±0.09 | **97.81±0.14** | 92.21±0.16 |
| Center-based Approximation | 200 | **94.92±0.03** | **95.66±0.06** | 97.80±0.02 | **92.24±0.04** |

*Table 8.* Convergence time in seconds (↓) using pairwise objective vs center-based approximation. The 1st (bold) results are highlighted to indicate their rankings.

| Settings | Cora | Citeseer | Pubmed | WikiCS |
|---|---|---|---|---|
| **Node Classification** | | | | |
| Pairwise Objective | ∼180 | ∼140 | ∼1500 | ∼1500 |
| Center-based Approximation | ∼**160** | ∼**110** | ∼**950** | ∼**1100** |
| **Link Prediction** | | | | |
| Pairwise Objective | ∼250 | ∼200 | ∼1800 | ∼450 |
| Center-based Approximation | ∼**220** | ∼**190** | ∼**1500** | ∼**400** |

GNN classifier better recognize nodes with overlapping embeddings. In addition, regardless of the boundary node selection strategy, the pre-update mechanism in Section 4.3 will help rescale potentially harmful large updates caused by mis-selected nodes under the contrastive objective. Therefore, we compute the shift score in the embedding space.

### D.2. Exact Pairwise Objective vs. Center-based Approximation

We initially attempted to optimize the exact pairwise contrastive loss (Thm. 3.1) by randomly sampling independent pairs. However, this yielded extensive computational overhead and chaotic convergence. Inspired by physical mean-field theories, we resolved this bottleneck by utilizing averaged group effects to approximate interactions, a strategy adopted by PCL (Li et al., 2021) from the contrastive learning domain. By operating on cluster centers instead of node instances, theoretical properties (identifiability, information bounds) established for individual pairs efficiently remain preserved through geometric identity, as described in Section 4.2. Empirically, this completely stabilizes convergence, accelerates training, and maximizes sample efficiency (Table 7 and Table 8).

### D.3. AUC and F1 value on Homogeneous Datasets

The comparison results of node classification on homogeneous datasets are shown in Table 9. Our method achieves the highest AUC and F1 scores on the majority of datasets, while maintaining stability and scalability, without the sharp performance declines observed in KGNN and SGNN. Standard deviations are the tightest (less than 0.35), ensuring reproducibility. Meanwhile, the comparison results of link prediction on homogeneous datasets are shown in Table 10. Our proposed method leads or matches every competitor on AUC and F1 value. The small performance standard deviations across all datasets of our method also demonstrate its robustness to graph size and domain shift.

### D.4. Performance on More Challenging Datasets

The evaluation results for node classification and link prediction of BES on more challenging graphs replicate robust gains, shown in Table 11 and Table 12. BES incorporates standard encoders augmented with a classifier, shown in Table 13. w/o BES represents baseline control runs fundamentally stripping out BES.

As shown in the node classification results, BES consistently achieves top-tier performance on both sets of datasets. These empirical results demonstrate that BES can effectively disentangle structural information in boundary regions, even in

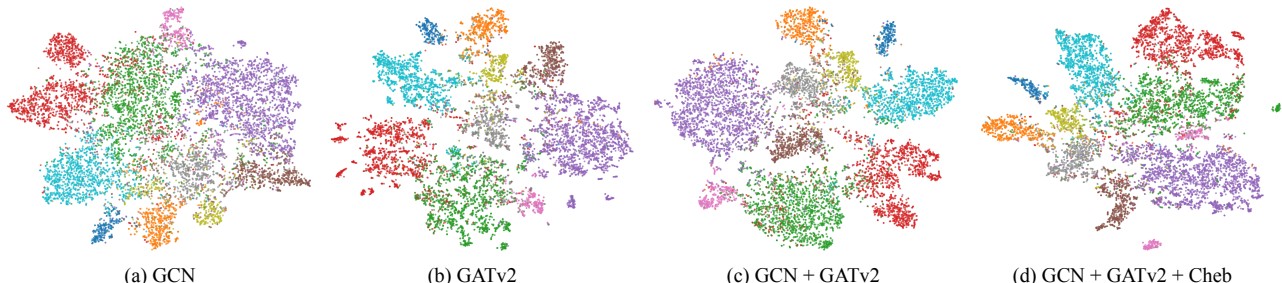

|     (a) GCN     |     (b) GATv2     |     (c) GCN + GATv2     |     (d) GCN + GATv2 + Cheb     |

*Figure 4.* T-SNE visualization of node embeddings generated by different graph encoders and their combinations.

*Table 9.* F1 (↑) and AUC (↑) of Node Classification on Different Datasets. OOM denotes out of memory. The 1st (bold) and 2nd (underlined) results are highlighted to indicate their rankings.

| | Cora | | CiteSeer | | Pubmed | | WikiCS | |
|---|---|---|---|---|---|---|---|---|
| Method | F1 | AUC | F1 | AUC | F1 | AUC | F1 | AUC |
| KGNN | 85.87±0.57 | 97.40±0.12 | 71.93±0.31 | 92.70±0.02 | 89.24±0.03 | 96.87±0.04 | 15.24±7.41 | 63.71±4.68 |
| GCN | 86.38±0.40 | 97.81±0.02 | 72.26±0.62 | 92.60±0.24 | 86.99±0.20 | 96.56±0.02 | 77.70±0.48 | 96.60±0.18 |
| Cheb | 85.95±0.69 | 97.74±0.05 | 71.03±1.12 | 92.39±0.29 | 89.61±0.15 | 97.37±0.03 | 79.77±0.33 | 97.24±0.05 |
| GAT | 85.76±0.34 | 97.91±0.33 | 72.45±0.91 | 92.92±0.27 | 86.42±0.16 | 96.35±0.13 | 77.85±0.58 | 97.07±0.09 |
| SGC | 86.91±0.50 | 97.86±0.12 | 74.10±1.06 | 94.25±0.11 | 87.37±0.10 | 96.53±0.03 | 77.06±0.85 | 96.48±0.21 |
| SSGC | 86.52±0.99 | 97.69±0.18 | **74.38±1.08** | 94.28±0.16 | 88.00±0.14 | 96.70±0.02 | 77.54±0.49 | 96.67±0.25 |
| GraphSAGE | 86.72±0.40 | 98.16±0.07 | 72.61±0.39 | 93.48±0.13 | 88.74±0.05 | 97.21±0.05 | 78.69±0.42 | 97.36±0.08 |
| GATv2 | 84.55±0.93 | 97.58±0.45 | 72.23±0.69 | 92.89±0.13 | 86.67±0.20 | 96.39±0.05 | 79.19±0.77 | 97.34±0.07 |
| H2GNN | 87.13±0.45 | 97.32±0.01 | 73.90±0.01 | 93.35±0.25 | 88.53±0.10 | 95.85±0.01 | 59.59±2.93 | 87.60±0.94 |
| SGNN | 86.27±0.07 | 91.68±0.03 | 74.14±0.74 | 84.51±0.37 | 83.12±0.19 | 87.27±0.20 | 65.47±2.51 | 79.88±0.96 |
| NodeImport-GCN | 77.32±0.36 | 95.95±0.34 | 61.35±1.22 | 86.45±1.38 | 83.56±0.24 | 95.15±0.07 | 73.05±0.09 | 95.49±0.09 |
| NodeImport-GAT | 76.15±0.52 | 94.41±0.53 | 58.25±0.96 | 85.97±1.45 | 83.56±0.32 | 95.16±0.07 | 75.37±0.24 | 95.86±0.08 |
| NodeImport-SAGE | 74.25±1.10 | 93.89±0.42 | 61.66±0.44 | 88.90±0.13 | 86.56±0.14 | 96.26±0.04 | 78.03±0.25 | 97.15±0.09 |
| MotifRGC-GCN | 86.37±0.66 | 97.71±0.26 | 72.47±0.31 | 91.99±0.18 | 88.41±0.17 | 96.95±0.01 | OOM | OOM |
| MotifRGC-GAT | 86.15±0.57 | 98.50±0.11 | 71.99±0.80 | 92.41±0.23 | 87.37±0.32 | 96.55±0.08 | OOM | OOM |
| MotifRGC-SAGE | 86.26±0.70 | 98.16±0.25 | 71.51±1.04 | 93.14±0.46 | 88.55±0.18 | 97.25±0.02 | OOM | OOM |
| MPNN-GCN | 83.61±0.34 | 95.85±0.05 | 72.79±0.43 | 93.53±0.13 | 89.50±0.10 | 97.49±0.01 | 82.44±0.27 | 97.49±0.03 |
| MPNN-GAT | 85.08±0.65 | 97.30±0.22 | 72.31±0.44 | 93.26±0.12 | 86.90±1.08 | 96.54±0.02 | 83.26±0.13 | 97.40±0.04 |
| MPNN-SAGE | 85.07±0.12 | 97.51±0.02 | 72.09±0.53 | 93.20±0.21 | 89.88±0.08 | 97.40±0.07 | 83.26±0.14 | 97.25±0.08 |
| GraphECL | 87.51±0.46 | 98.44±0.00 | 70.72±1.46 | 93.25±0.88 | 88.05±0.05 | 96.93±0.03 | 79.09±0.07 | 97.26±0.02 |
| IGCL | 87.29±0.88 | **98.53±0.09** | 73.80±0.27 | **94.65±0.03** | 87.08±0.04 | 96.72±0.01 | 80.52±0.07 | 97.37±0.01 |
| OURS | **88.41±0.35** | 97.65±0.15 | 74.21±0.05 | 92.86±0.04 | **90.92±0.02** | **97.57±0.02** | **83.36±0.01** | **97.66±0.09** |

complex real-world scenarios where the theoretical prerequisite might be partially relaxed.

## D.5. Embeddings from Different Graph Encoders

We aim to present the features generated by various graph encoders, as each encoder captures and represents the data from a distinct perspective. The produced node representation in the embeddings space of nodes from different graph encoders and their combinations is shown in Figure 4.

We use GCN and GATv2 graph encoders in the experiments. The embedding visualization results reveal that the raw embeddings produced by individual graph encoders, including GCN, GATv2, and their combinations, exhibit significant category overlap. However, the combined use of multiple graph encoders demonstrates enhanced feature extraction capability, facilitating more effective classification when employing the Cheb classifier (Defferrard et al., 2016), which leverages diverse feature representations derived from different perspectives of the data.

*Table 10.* F1 (↑) and AUC (↑) of Link Prediction on Different Datasets. - sign indicates the method does not support the evaluation. OOM denotes out of memory. The 1st (bold) and 2nd (underlined) results are highlighted to indicate their rankings.

| Method | Cora | | CiteSeer | | Pubmed | | WikiCS | |
|---|---|---|---|---|---|---|---|---|
| | F1 | AUC | F1 | AUC | F1 | AUC | F1 | AUC |
| KGNN | 85.58±0.53 | 91.61±0.09 | 88.21±0.49 | 95.35±0.31 | 85.59±0.14 | 93.56±0.18 | – | – |
| GCN | 92.58±0.22 | 96.45±0.23 | 95.41±0.31 | 98.75±0.05 | 93.80±0.22 | 97.87±0.04 | 91.92±0.12 | 97.26±0.07 |
| Cheb | 90.96±0.32 | 95.00±0.22 | 89.82±0.21 | 95.47±0.13 | 87.08±0.04 | 94.64±0.04 | 90.95±0.13 | 96.73±0.07 |
| GAT | 90.66±0.18 | 96.54±0.31 | 92.42±0.36 | 97.81±0.26 | 89.52±0.10 | 95.30±0.06 | 90.94±0.10 | 96.52±0.06 |
| SGC | 91.91±0.18 | 96.59±0.15 | 94.43±0.17 | 98.52±0.09 | 86.43±0.02 | 95.10±0.03 | 92.05±0.14 | **97.35±0.01** |
| SSGC | 91.72±0.13 | 96.43±0.09 | 94.38±0.35 | 98.40±0.17 | 89.04±0.16 | 96.05±0.08 | 91.63±0.30 | 97.26±0.13 |
| GraphSAGE | 91.04±0.19 | 95.79±0.25 | 91.96±0.28 | 96.73±0.12 | 88.68±0.13 | 94.69±0.07 | 90.23±0.22 | 95.79±0.11 |
| GATv2 | 89.77±0.21 | 95.63±0.25 | 93.22±0.14 | 97.84±0.08 | 88.55±0.15 | 94.71±0.08 | 90.19±0.18 | 96.08±0.07 |
| H2GNN | 92.47±0.48 | 96.50±0.12 | 95.05±0.24 | 98.30±0.09 | 95.79±0.10 | 98.81±0.05 | 92.12±0.04 | 97.32±0.07 |
| SGNN | 93.61±0.54 | 97.60±0.12 | 96.12±0.37 | 98.23±0.03 | 94.72±0.10 | 98.35±0.04 | 91.02±0.20 | 96.85±0.08 |
| NodeImport-GCN | 87.66±0.40 | 92.69±0.32 | 85.85±0.23 | 91.46±0.21 | 86.03±0.03 | 92.42±0.06 | 89.73±0.12 | 96.01±0.16 |
| NodeImport-GAT | 87.47±0.73 | 92.96±0.13 | 83.03±0.27 | 89.99±0.16 | 85.66±0.03 | 91.98±0.19 | 89.47±0.18 | 95.92±0.04 |
| NodeImport-SAGE | 80.29±1.32 | 87.45±0.14 | 77.48±0.35 | 85.68±0.17 | 79.25±0.15 | 85.93±0.05 | 86.43±0.09 | 93.37±0.11 |
| MotifRGC-GCN | 72.51±1.51 | 80.14±0.73 | 86.85±1.53 | 92.80±0.94 | 93.51±0.13 | 97.87±0.04 | OOM | OOM |
| MotifRGC-GAT | 88.55±0.28 | 93.50±0.11 | 87.66±2.96 | 92.93±1.82 | 97.64±0.06 | 99.58±0.02 | OOM | OOM |
| MotifRGC-SAGE | 89.03±0.58 | 94.36±0.56 | 91.24±1.55 | 95.75±0.65 | 97.52±0.05 | 99.44±0.10 | OOM | OOM |
| MPNN-GCN | 91.28±0.32 | 96.46±0.17 | 93.96±0.15 | 98.22±0.15 | 86.46±0.12 | 94.68±0.03 | 88.74±0.09 | 95.83±0.10 |
| MPNN-GAT | 89.82±0.48 | 94.78±0.11 | 90.46±0.04 | 95.43±0.28 | 86.26±0.14 | 92.61±0.10 | 85.84±0.21 | 92.87±0.16 |
| MPNN-SAGE | 88.45±0.19 | 92.75±0.17 | 90.27±0.16 | 95.63±0.08 | 81.80±0.06 | 85.32±0.05 | 84.72±0.06 | 91.62±0.14 |
| GraphECL | 92.84±0.25 | 97.08±0.09 | 95.26±0.24 | 98.68±0.06 | 94.19±0.05 | 98.00±0.05 | 11.07±15.66 | 52.16±3.06 |
| IGCL | **95.13±0.37** | **99.15±0.16** | 95.67±0.15 | 98.56±0.21 | 88.81±0.09 | 94.48±0.02 | 89.84±0.14 | 96.13±0.06 |
| OURS | 94.45±0.03 | 99.03±0.01 | **96.26±0.08** | **99.27±0.04** | **97.89±0.01** | **99.68±0.02** | **92.27±0.05** | 97.03±0.07 |

## D.6. Visualization of the Boundary Embedding Shaping

To show the effectiveness of the boundary embedding shaping process, we use binary and multiclass node classification scenarios for the analysis, shown in Figure 5. The solid line denotes the decision hyperplane projected onto the two-dimensional embedding space, while the dashed lines indicate the corresponding boundary region, from which boundary nodes are conditionally selected to guide the refinement of node embeddings. From the binary shaping results, we can observe that the boundary regions between node embeddings from two classes are clarified after the boundary embedding shaping process without causing large embedding changes of other nodes. The same trend can also be observed for multi-class shaping.

The boundary embedding shaping on Cora and CiteSeer is shown in Figure 6 and Figure 7. From the results, we can see that the node embeddings in the boundary regions are also clarified after the boundary embedding shaping workflow, which improves the final classification performance on the corresponding datasets.

## D.7. Ablation Studies

We evaluate the impact of different model components on overall accuracy, taking both node classification and link prediction as examples, shown in Table 14 and Table 15, respectively. The results indicate that simply concatenating multiple encoders does not always outperform a single encoder. In contrast, when BES is integrated, the model consistently yields better performance compared to both single and combined encoder settings. We believe this serves as strong evidence that BES effectively enhances performance by optimizing the representations of boundary nodes.

Specifically, these results illustrate an average performance gain of more than 1% when BES is applied to the boundary region on WikiCS. By comparison with the single encoder setting, the performance gain is even more significant, where a 2-3% performance gain is achieved. This trend is consistently observed across other datasets as well, where a 1-2% performance gain is achieved. The node classification accuracy achieved in Layer 1+2 surpasses that of Layer 1 on most datasets, such as Cora, Citeseer, and WikiCS. A similar trend of accuracy enhancement is observed in the link prediction task, where the combined Layer 1+2 consistently outperforms Layer 1 on most datasets. Furthermore, the variance in

*Table 11.* Node classification accuracy (↑) on heterogeneous, heterophilic and OGB graphs. OOM indicates out of memory. The 1st (bold) results are highlighted to indicate their rankings.

| Methods | Chameleon | Cornell | Texas | Roman-empire | ogbn-arxiv |
|---|---|---|---|---|---|
| KGNN | 50.55 | **89.19** | 78.38 | 71.45 | 45.55 |
| GCN | 45.27 | 43.24 | 62.16 | 41.09 | 71.62 |
| Cheb | 43.52 | 81.08 | 78.38 | 74.18 | 71.95 |
| GAT | 49.45 | 51.35 | 64.84 | 42.98 | 71.93 |
| SGC | 40.66 | 45.95 | 59.46 | 42.56 | 67.01 |
| SSGC | 40.64 | 43.24 | 59.46 | 42.89 | 67.20 |
| GraphSAGE | 52.97 | 83.78 | 86.49 | 69.26 | 71.61 |
| GATv2 | 47.69 | 48.65 | 59.46 | 63.79 | 72.25 |
| MotifRGC-GCN | 37.36 | 33.51 | 47.57 | 44.15 | OOM |
| MotifRGC-SAGE | 54.21 | 78.38 | 76.76 | 57.61 | OOM |
| MotifRGC-GAT | 46.29 | 42.65 | 49.19 | 49.41 | OOM |
| MPNN-GCN | 49.67 | 40.54 | 54.05 | 37.61 | 66.22 |
| MPNN-SAGE | 53.41 | 51.35 | 83.78 | 74.42 | 65.28 |
| MPNN-GAT | 47.25 | 55.86 | 81.08 | **81.87** | 69.17 |
| IGCL | 49.01 | 81.08 | 89.10 | 66.31 | 71.59 |
| w/o BES | 53.41 | 81.08 | 83.78 | 80.87 | 73.05 |
| BES(Ours) | **54.29** | **89.19** | **89.19** | 81.47 | **75.04** |

*Table 12.* Link prediction accuracy (↑) on OGB graphs. OOM indicates out of memory. The 1st (bold) results are highlighted to indicate their rankings.

| Methods | ogbl-collab |
|---|---|
| KGNN | 64.77 |
| GCN | 64.21 |
| Cheb | 64.77 |
| GAT | 64.91 |
| SGC | 62.40 |
| SSGC | 59.99 |
| GraphSAGE | 62.68 |
| GATv2 | 64.30 |
| w/o BES | 62.68 |
| BES(Ours) | **65.84** |

performance is reduced across all datasets following the application of boundary embedding shaping. That means both layers learn the local node features while preserving the relevant topological information. In addition, the joint application of multiple loss functions during different layers of training not only mitigates the limitations of individual end-to-end loss functions but also enhances the model's discriminative power through cooperative optimization mechanisms, thereby confirming its effectiveness in handling complex graph data.

The boundary node is crucial, and we also provide an ablation study with and without the boundary node to compute accuracy across different datasets in Table 16. Results confirm consistent gains with boundary nodes.

### D.8. Convergence and Efficiency Analysis

We evaluate the convergence and efficiency of the proposed method on different datasets during the training process for node classification and link prediction, shown in Figure 8 and Figure 9, respectively. The node classification results indicate that, on the Cora and Citeseer datasets, the proposed method achieves convergence within 100 seconds for both Layer 1 and the combined Layer 1+2. Meanwhile, on the PubMed and WikiCS datasets, convergence is attained within 1,000 seconds for the corresponding layers. Meanwhile, the results of the link prediction on the Cora and Citeseer datasets show that the proposed method reaches convergence within 200 seconds for different layers. Subsequently, on the WikiCS dataset, convergence is also attained around 400 seconds. On the PubMed dataset, the training exhibits initial fluctuations, with convergence observed after approximately 1,500 seconds.

*Table 13.* Encoders and Classifier Configuration for BES on Different Datasets

|  | Chameleon | Cornell | Texas | Roman-empire | ogbn-arxiv | ogbl-collab |
|---|---|---|---|---|---|---|
| Encs | GraphSAGE+KGNN | Cheb+KGNN | Cheb+GraphSAGE | Cheb+KGNN | Cheb+GATv2 | GraphSAGE |
| Cls | GraphSAGE | Cheb | Cheb | Cheb | Cheb | DOT |

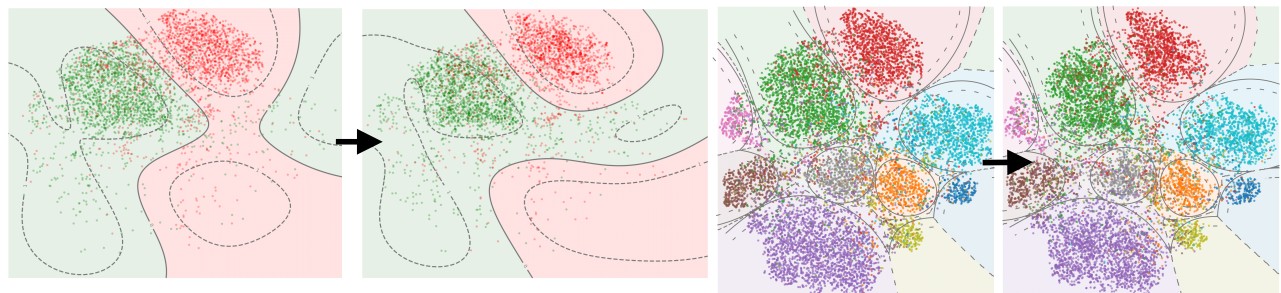

(a) Before/After Boundary Embedding Shaping (Binary class)  (b) Before/After Boundary Embedding Shaping (Multi-class)

*Figure 5.* Boundary embedding shaping. (a) is for the binary scenario; (b) is for the multi-class scenario.

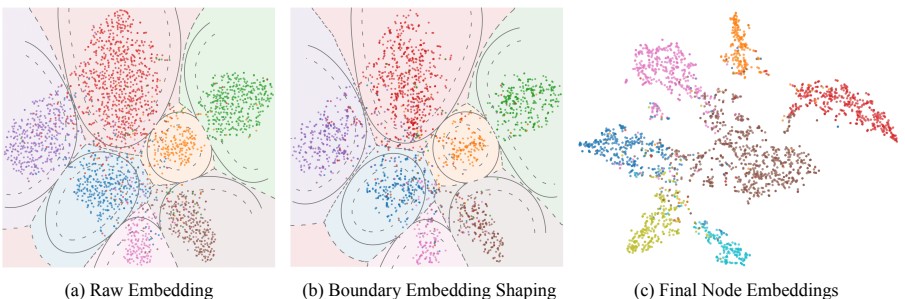

(a) Raw Embedding  (b) Boundary Embedding Shaping  (c) Final Node Embeddings

*Figure 6.* Boundary embedding shaping for Cora.

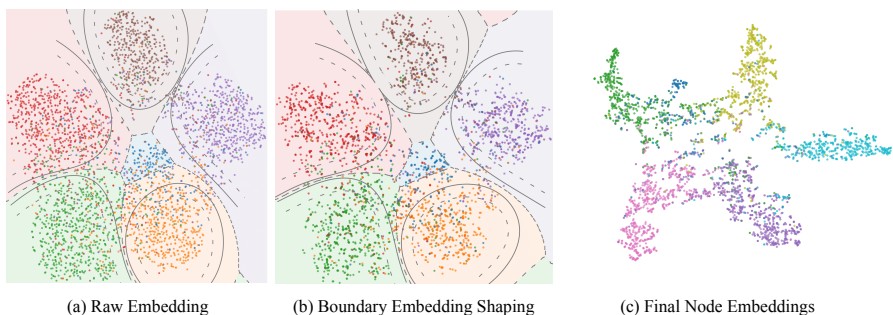

(a) Raw Embedding  (b) Boundary Embedding Shaping  (c) Final Node Embeddings

*Figure 7.* Boundary embedding shaping for CiteSeer.

*Table 14.* Ablation studies of BES for node classification on accuracy ($\uparrow$). The 1st (bold) results are highlighted to indicate their rankings.

| Settings | Enc1 | Enc2 | Cls | L1 | L1+2 | Cora | Citeseer | Pubmed | WikiCS |
|---|---|---|---|---|---|---|---|---|---|
| Enc1 | ✓ |  | Cheb |  |  | 87.31±0.76 | 75.74±0.74 | 89.80±0.14 | 82.52±0.32 |
| Enc2 |  | ✓ | Cheb |  |  | 87.74±0.44 | 78.10±0.63 | 89.35±0.02 | 81.71±0.69 |
| Enc1+2 | ✓ | ✓ | Cheb |  |  | 88.55±0.10 | 78.00±0.20 | 90.30±0.017 | 84.21±0.25 |
| L1 | ✓ | ✓ | Cheb | ✓ |  | 89.40±0.17 | 78.20±0.06 | **91.03±0.03** | 85.30±0.15 |
| L1+2 | ✓ | ✓ | Cheb | ✓ | ✓ | **89.46±0.15** | **78.40±0.07** | 90.92±0.07 | **85.52±0.08** |

*Table 15.* Ablation studies of BES for link prediction on accuracy (↑). The 1st (bold) results are highlighted to indicate their rankings.

| Settings | Enc | Cls | L1 | L1+2 | Cora | Citeseer | Pubmed | WikiCS |
|---|---|---|---|---|---|---|---|---|
| Enc | ✓ | Cheb | | | 92.98±0.08 | 95.60±0.18 | 97.64±0.06 | 92.00±0.15 |
| L1 | ✓ | Cheb | ✓ | | **94.92±0.03** | 95.66±0.06 | 97.80±0.02 | 92.24±0.04 |
| L1+2 | ✓ | Cheb | ✓ | ✓ | 94.45±0.03 | **96.27±0.04** | **97.89±0.01** | **92.27±0.05** |

*Table 16.* Accuracy (↑) for node classification with **w/** and **w/o** boundary nodes. The 1st (bold) results are highlighted to indicate their rankings.

| Settings | Cora | Citeseer | Pubmed | WikiCS | Chameleon | Cornell | Texas | Roman-empire | ogbn-arxiv |
|---|---|---|---|---|---|---|---|---|---|
| w/o boundary nodes | 90.02 | 78.10 | 90.59 | 83.97 | 53.85 | 81.08 | **89.19** | 80.96 | 72.54 |
| w/ boundary nodes | **90.39** | **78.40** | **91.03** | **85.52** | **54.29** | **81.19** | **89.19** | **81.47** | **75.04** |

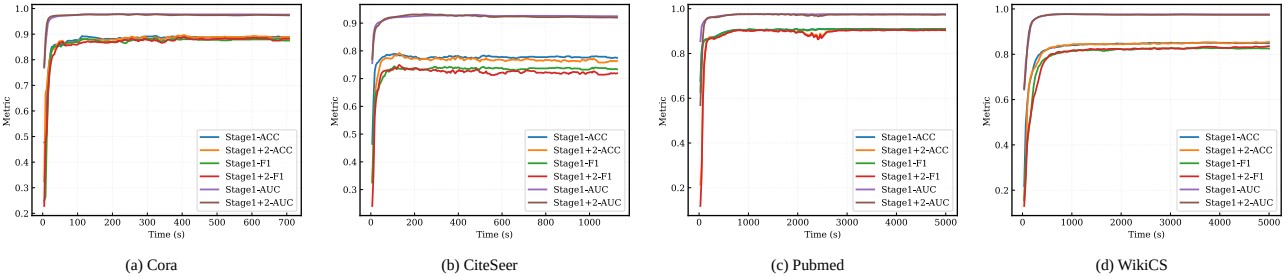

(a) Cora     (b) CiteSeer     (c) Pubmed     (d) WikiCS

*Figure 8.* Metric convergence of node classification on different datasets during training.

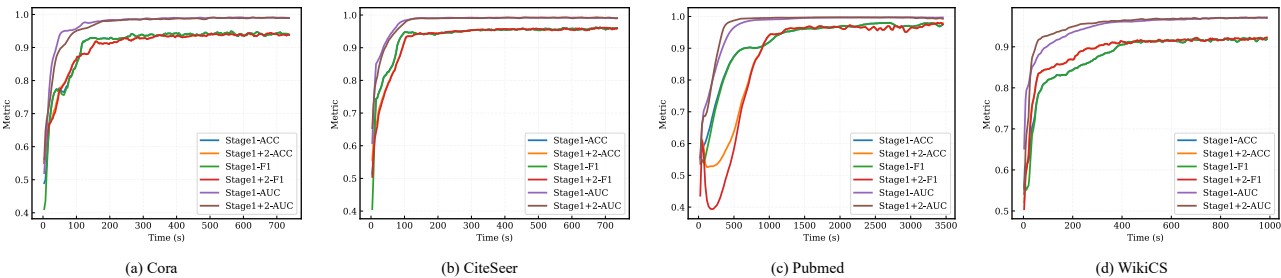

(a) Cora     (b) CiteSeer     (c) Pubmed     (d) WikiCS

*Figure 9.* Metric convergence of link prediction on different datasets during training.

