# OpenReview forum: "Boundary Embedding Shaping with Adaptive Contrastive Learning for Graph Structural Disentanglement"
_ICML.cc/2026/Conference — ICML 2026 regular_

### Official Review · Reviewer_Szpd · 2026-03-09

**Soundness:** 3
**Presentation:** 3
**Significance:** 3
**Originality:** 2
**Overall Recommendation:** 5
**Confidence:** 3

**Summary:**

This paper proposes the Boundary Embedding Shaping (BES), a method to improve the classification performance by selectively suppressing spurious structural noise at decision boundaries. Their boundary-focused contrastive methods improve the embeddings with gravity loss, and are verified on several datasets and link prediction benchmarks. The paper also provides interpretability analyses and ablation studies to support the proposed method.

**Compliance With Llm Reviewing Policy:**

Affirmed.

**Final Justification:**

The theoretical proof established by the author makes my agree more on their method. I think this paper should be considered more to get accept.

**Key Questions For Authors:**

1 In Algorithm 1, the method iterates over each layer and performs several update steps per layer. Could the authors clarify the time complexity introduced by the Boundary Embedding Shaping procedure relative to the base GNN encoder?

2 Does the method require label information during training? It appears that labels may be used to identify positive pairs in the contrastive learning objective. If so, the method would not be fully self-supervised. Could the authors clarify how positive pairs are constructed in practice and explain “we connect the two nodes form different local sub-graphs in a positive pair” further to me in line 147.

**Limitations:**

yes

**Strengths And Weaknesses:**

Soundness.

Good. The proposed intuition is reasonable, and the theoretical analysis supporting the method appears sound. The empirical results further validate the approach.


Presentation:

Good. The overall narrative is clear, and the paper is generally well structured. The interpretability analysis is helpful in understanding the behavior of the method, although some of the intuitions could be explained more clearly.


Significance:

Good. It seems that these methods can improve the classification accuracy  on several benchmarks.


Originality:

Fair. The idea of leveraging label information or boundary-aware signals to refine embeddings is not entirely new. However, the proposed framework provides useful insights into how contrastive learning can be applied to mitigate graph structural entanglement.



Weakness:

1 The paper does not provide a clear analysis of the computational or time complexity of the proposed method. Since the Boundary Embedding Shaping module is applied after the GNN encoder and involves additional embedding refinement steps, it would be helpful to understand the additional computational overhead introduced by this procedure.

---

> ### Author Rebuttal · Authors · 2026-03-30
>
> We deeply value the careful reading and highly constructive validation logic suggestions.
>
> > A. W (1) and Q (1): Complexity Analysis
>
> Let $N$ be the number of nodes, $D$ be the embedding dimension, $C$ be the number of classes, $K$ be the average node degree, and $Q$ be the number of iterations per layer $L$:
>
> **Time Complexity:** BES operations within each layer include (1) Statistical Parameter Estimations natively tracking $\mathcal{O}(ND^2+CD^3)$; (2) Topological Boundary Tracing locally bounding $\mathcal{O}(NCD)$; (3) Positional Distance Queries bounded tightly by $\mathcal{O}(ND\log N)$. (4) compute global perturbations $\Delta^{(B)}$, the core computation bottleneck of BES, for attention training on all $N$ nodes, taking $\mathcal{O}(N K D^2)$ time for each of the $Q$ iterations. Thus, the overall time complexity is $\mathcal{O}(L\cdot QNKD^2)$.
>
> **Space Complexity:** BES explicitly caches the node embeddings $\mathcal{O}(N D)$, the sparse graph adjacency $\mathcal{O}(N K)$, the Gaussian distribution parameters $\mathcal{O}(C D^2)$, and the attention layer weights $\mathcal{O}(D^2)$. Therefore, the space complexity is strictly bounded at $\mathcal{O}(ND + NK + CD^2)$, rendering the algorithm highly memory-efficient and comparable to standard message-passing mechanisms.
>
> To address the $\mathcal{O}(N)$ scalability bottleneck, we further provide practical acceleration strategies. Specifically, the node-independent attention operations in $\Delta^{(B)}$ can be reformulated as Sparse-Dense Matrix Multiplications (SpMM), enabling efficient GPU parallelization via modern sparse tensor backends. In addition, for large-scale graphs, $\Delta^{(B)}$ can be computed using multi-processing or distributed node partitioning under a Map-Reduce paradigm, where the graph is horizontally partitioned to reduce synchronization overhead. These deployment-ready strategies effectively ensure both computational efficiency and scalability in practice.
>
>
> > B. Q (2): Label-informed Training
>
> We clarify that BES is not a fully self-supervised method, but an embedding refinement framework that explicitly leverages training label information from training nodes (while strictly excluding all test nodes and test labels). These training labels are used in a targeted manner to identify nodes near the decision boundary and guide the attention mechanism to suppress structural noise.
>
> Label guidance is critical in achieving structural disentanglement. It helps the model effectively isolate noisy neighbors, reduce feature entanglement, and learn more reliable intrinsic node representations. In contrast, a purely self-supervised approach (e.g., based only on feature similarity) would inevitably treat heterophilic noise as intrinsic features, exacerbating structural entanglement and making it difficult for the downstream classifier to distinguish useful signals from noise. Therefore, label-informed training is essential for accurate topological noise filtering.
>
> The pair construction described in line 147 shows that a positive pair refers to nodes of the same class but from different local topological contexts, while a negative pair consists of nodes from different classes. In practice, we approximate this pair-wise process using cluster centers derived from labeled training nodes. Each node is pulled toward its class center and pushed away from others, improving sample efficiency and training stability.
>
>
> > C. Connection between BES and existing CL methods and originality concerns
>
> We acknowledge that class-guided contrastive learning is a well-established paradigm, as demonstrated by SCL and PCL, and adopted in recent graph frameworks (e.g., GraphECL, POT).
> At a high level, BES shares common ground with SCL, PCL, and SwAV in avoiding exhaustive pairwise comparisons. Instead, it employs cluster centers (prototypes) as anchors to approximate global distributions, ensuring computational efficiency and optimization stability. However, BES introduces three key adaptations tailored to graph-structured data:
>
> 1. Intrinsic Motivation:
> SCL and PCL are designed for i.i.d. data (e.g., images) and focus solely on feature alignment. In contrast, BES targets structural entanglement in graphs, where message passing mixes heterophilic neighbors. BES is not just for feature clustering, but a mechanism to suppress noisy topological neighbors.
>
> 1. Update Mechanism:
> SCL and PCL apply uniform updates to all samples. In dense graphs, this can lead to representation collapse and obscure disentanglement. BES instead introduces an adaptive pre-update mechanism that selectively updates nodes that can be disentangled from their noise neighbors, leaving confident interior nodes unchanged.
>
> 1. Optimization Dynamics:
> Rather than relying on standard gradient updates, BES computes global boundary perturbations ($\Delta^{(B)}$) to dynamically scale the learning step size. This ensures that local boundary refinement does not disrupt global graph representations.

---

> > ### Author Rebuttal · Reviewer_Szpd · 2026-04-02
> >
> > Thank you for the details. My concerns have been fully addressed.

---

> > > ### Author Response · Authors · 2026-04-05
> > >
> > > Thank you for acknowledging our rebuttal and for carefully reviewing both our responses and those of other reviewers.
> > > We are glad that the clarifications and additional analysis addressed your concerns.
> > > Building upon our earlier revisions, we have now formalized a strictly deductive mathematical bridge to establish a **much tighter connection** between our theory and the proposed method, ensuring the overall presentation is significantly more rigorous and clear (please refer to our detailed responses to Reviewer **c6fM** and Reviewer **vPC5**).
> > > We will incorporate all of these updates into the revised version.
> > > Thank you again for helping us improve the quality of our paper.

---

### Official Review · Reviewer_vPC5 · 2026-03-11

**Soundness:** 2
**Presentation:** 3
**Significance:** 2
**Originality:** 3
**Overall Recommendation:** 3
**Confidence:** 3

**Summary:**

This paper studies graph structural entanglement in GNNs, with a focus on nodes near class boundaries where noisy neighborhood information can make classification harder. To address this issue, the paper proposes Boundary Embedding Shaping (BES), a plug-in module that first selects boundary nodes, then uses a contrastive objective to pull these nodes toward class centers, and finally uses an adaptive update rule to limit parameter perturbation. The paper also gives a theoretical section based on invariant and variant latent factors, and reports experiments on node classification and link prediction showing that BES improves over several baselines on four benchmark datasets.

**Compliance With Llm Reviewing Policy:**

Affirmed.

**Final Justification:**

The rebuttal and follow-up improve the paper, but they still do not fully close the main concern: the theory is built around a generic contrastive setting, while the practical BES pipeline adds boundary-node selection, Gaussian class centers, gravity loss, and adaptive update scaling, so I am still not convinced the actual method is tightly derived from the theorem; the added experiments help, but because this core gap remains and the limitations are still material, I would maintain my score.

**Key Questions For Authors:**

1. What labels are used when computing the boundary-node shift score and the class centers? Please state clearly whether BES uses only training labels, pseudo-labels, or labels for all nodes.

2. Can the authors clarify the exact connection between the theoretical results and the actual BES objective? In particular, how does the theorem for generic contrastive learning with positive pairs translate to the boundary-node and class-center gravity loss used in practice?

3. Can the authors provide stronger experiments on more challenging graph settings? For example, larger datasets, stronger heterophily, or noisier neighborhood structure would help show whether BES generalizes beyond the current benchmarks.

4. Can the authors include a cleaner ablation that isolates the contribution of each part of BES? I would especially like to see separate effects of boundary-node selection, gravity loss, and the adaptive update mechanism.

5. How much additional computation does BES add compared with the strongest baselines under matched backbone settings?

**Limitations:**

No. The paper does not discuss some important limitations clearly enough. In particular, it should discuss the strong assumptions behind the theory and boundary modeling, the unclear label usage in boundary-node selection, the limited benchmark diversity, and the fact that the theory does not map very directly to the exact practical objective.

**Strengths And Weaknesses:**

Strengths

1. The paper studies a meaningful problem: Graph structural noise and boundary-node difficulty are real issues in graph learning, and the paper gives a clear motivation for focusing on this part of the embedding space.

2. The idea of selecting hard boundary nodes and refining them with a targeted contrastive objective is intuitive. The adaptive update rule is also a reasonable design choice to avoid changing all embeddings too much.

3. The paper reports both node classification and link prediction results, compares against many classical and recent baselines, and shows consistent gains on Cora, CiteSeer, Pubmed, and WikiCS.

4. The paper is generally organized well. The overall flow from motivation to method to experiments is easy to follow, and the figures help explain the main idea.

Weakness:

1. The theoretical part proves an identifiability result for a generic contrastive setting with strong assumptions, but the practical BES method uses boundary-node selection, class centers, and a special gravity loss. I do not think the paper clearly shows that the theorem really justifies the actual training procedure.

2. The use of labels in boundary-node selection is unclear and potentially problematic. The shift score uses whether neighboring nodes have different labels, and the method also uses class centers. In a semi-supervised node classification setting, it is very important to clarify whether the method uses only training labels or labels of all nodes. If it uses all labels, that would be a serious issue. If it uses only training labels or pseudo-labels, the implementation needs to explain this clearly.

3. The assumptions behind the method look fairly strong. The boundary region is built with Gaussian cluster assumptions and linear separability, and the method seems to depend on having reliable class-centered structure in the embedding space. I am not fully convinced this will hold in harder or noisier graph settings.

4. The experimental setup is still somewhat limited. The paper uses four standard citation-style datasets, which are useful but relatively small and common. I would like to see stronger evidence on larger, more diverse, or more challenging graphs, especially settings with stronger heterophily or noisier structure.

5. The comparison story is not fully clean. BES is a plug-in refinement module added on top of existing encoders, while many baselines are standalone methods. Because of this, it is a bit hard to judge whether the gain mainly comes from the boundary-shaping idea itself or from extra modeling flexibility and extra computation.

6. Some claims are written too strongly. The paper often presents structural disentanglement and theoretical justification in a very strong way, but the current evidence feels more supportive than conclusive.

---

> ### Author Rebuttal · Authors · 2026-03-30
>
> We sincerely appreciate the constructive and insightful comments.
>
> > A. W (1) and Q (2): Connection between Theory and Method
>
> Please refer to Response A of Reviewer c6fM.
>
> > B. W (2) and Q (1): Details of the label utilization
>
> Please refer to Response B of Reviewer Szpd.
>
> > C. W (3): Idealized conditions and assumptions
>
> We now fully agree that claims such as “linear separability” should be relaxed to better align with the method’s potential and new empirical results, and we also thoroughly understand the concern regarding the gap between theoretical idealizations and real-world graph noise. However, we respectfully clarify that establishing these restrictive assumptions is a necessary "first-principles" foundation. In the chaotic domain of structural entanglement, providing a trustworthy mathematically provable baseline under representative cases is essential; otherwise, subsequent heuristic designs lack formal guarantees.
>
> Crucially, our practical algorithm does not always assumes real-world graphs is perfect. BES incorporates extraordinarily robust algorithmic relaxations:
>
> 1. Dynamic Boundary Tracing: Instead of enforcing strict linear hyperplanes, BES utilizes localized topological shift scores to dynamically map highly irregular, non-Gaussian decision boundaries.
> 2. Adaptive Perturbation Scaling ($\Delta^{(B)}$): To proactively counteract extreme topological noise or missing edges, our pre-update mechanism strictly bounds gradient traversals, safely mitigating unforeseen structural corruptions.
>
> Empirical Verification:
> The final validation of these relaxations is empirical. As detailed in Tab B.3-B.5 (Response E of Reviewer J2Py), we aggressively challenged BES across exceptionally chaotic, non-ideal environments: fragmented heterogeneous datasets (Chameleon, Cornell, Texas), severe heterophilic manifolds (Roman-empire), and massive industrial benchmarks featuring millions of noisy edges (ogb-arxiv and ogb-collab).
>
> Across these overwhelmingly harsh scenarios, BES consistently achieves top-tier performance. This clear empirical evidence conclusively proves that our restrictive foundational theory successfully captures the core mechanics of structural entanglement, while our deliberate algorithmic relaxations universally empower BES to thrive in complex, real-world graphs.
>
> > D. W (4) and Q (3): Additional experiments on harder datasets
>
> Please refer to Response E of Reviewer J2Py.
>
> > E. W (5) and Q (4): Provide cleaner ablation experiments
>
> Please refer to Response B of Reviewer c6fM.
>
> > F. W (6): strong claims
>
> Please refer to Response C above.
>
> > G. Q (6): Computation overhead of BES
>
> Please refer to Response A of Reviewer Szpd.
>
> > H. For the lack of a Limitation section
>
> We appreciate the reviewer for pointing this out. Before detailing the limitations, we briefly restate the core intuition of our method to contextualize its boundaries:
>
> 1. In real-world graphs, factors such as structural noise and missing edges often cause the neighborhoods of nodes from different classes to intermingle.
> 2. Traditional message-passing mechanisms are highly vulnerable to this structural noise, resulting in mixed-up node embeddings that fail to reflect the true intrinsic features of the nodes.
> 3. To address this, we leverage class information to select specific nodes and modify their embeddings through BES, thereby weakening the effect of neighbor noise. The less mixed-up the embeddings are, the weaker the noise becomes. This makes the node representations much more reliable in revealing their true features.
> 4. Finally, the classifier utilizes these reliable features to recognize and filter out noisy neighbors, effectively achieving noise filtering via structural disentanglement.
>
> Given this mechanism, we acknowledge that BES has limitations. As an early attempt to explicitly address this structural entanglement problem—which remains an ongoing challenge in the community—we focused on establishing a solution for the most foundational cases, guided by relatively strict assumptions.
>
> Currently, BES is primarily designed by utilizing class-guided signals, meaning its effectiveness relies on having sufficient labeled nodes to accurately estimate the representation distribution of different categories. Furthermore, while BES excels at separating overlapping representations in boundary regions, it may struggle in extreme cases where the initial latent representations of different categories are severely overlapped, with the center of the distribution being the same or lacking any discriminative initial features (such as overwhelmingly strong heterophily without any homophilous clues).
>
> We view these constraints as promising avenues for future work, where this foundational framework can be relaxed and extended to handle more complex or extreme distributions.

---

> > ### Author Rebuttal · Reviewer_vPC5 · 2026-04-04
> >
> > Thanks for your reply. This rebuttal improves the paper, but it still does not fully close the main concerns. The strongest remaining gap is still the theory-to-method connection: the paper claims that the theory motivates BES, but the formal result is built on a label-informed positive-pair assumption and a generic contrastive objective, while the actual method adds several specific design choices such as boundary-node selection with class labels, Gaussian class centers, gravity loss, and adaptive update scaling, so my concern that the practical BES pipeline is not tightly derived from the theorem remains largely valid. Besides, the paper still has limitations, as the author acknowledged. As a result, I would like to maintain my score.

---

> > > ### Author Response · Authors · 2026-04-05
> > >
> > > We sincerely thank you for your continued engagement and deeply value the high bar you set for theoretical rigor. We fully acknowledge that the previous rebuttal left the theory-to-method connection insufficiently formalized. Guided directly by your feedback, we have rewritten Sections 3 and 4 to establish a step-by-step mathematical bridge, so that every design choice in BES has no heuristic gaps. Below are the key revisions.
> > >
> > > **Remark 3.1 (Section 3): connecting assumption to graph heterophily.**
> > > The paper addresses the structural entanglement from the graph heterophily, so Remark 3.1 for Asm 3.2 restates this as: *"Positive sample pairs sharing the same semantic class ($\mathbf{z}\_{inv}$) frequently connect to structurally diverse neighborhoods, causing natural variations in $\mathbf{z}\_{var}$. Our method turns this into an advantage: it natively fulfills the sufficient variation mandate of Asm 3.2."* The theory is thus not generic but inherently tied to graph topology, and Asm 3.2 is satisfied by boundary node selection with shift score (training-label-informed) for later structural disentanglement.
> > >
> > > **Section 4 Reorganization: three deductively theory-grounded steps.**
> > > We reorganized Section 4 to close each gap you identified:
> > > - 4.1 deriving the pairwise gravity loss directly from Thm. 3.1
> > > - 4.2 proving that it is gradient-equivalent to the center-based objective via an algebraic identity for approximation
> > > - 4.3 motivating adaptive scaling from Prop. 3.1's error bound
> > >
> > > **4.1 Pairwise Contrastive Objective**: Boundary node selection explicitly identifies nodes satisfying Asm. 3.2 via the label-informed shift score $S$ computation (high heterophily = sufficient structural variation). For a boundary node $v\_b$, the pairwise gravity similarity terms directly instantiate $\mathrm{sim}(\cdot,\cdot)$ in Thm. 3.1, positive gravity pulls same-class nodes, negative gravity pushes different-class nodes:
> > > $$\mathrm{sim}^{pos}\_{g}(\Phi(\mathbf{x}\_b), \Phi(\mathbf{x}\_p)) = -||\Phi(\mathbf{x}\_b) - \Phi(\mathbf{x}\_p)||\_2^2,\; v\_p \in \mathcal{C}\_{y\_b},$$
> > > $$\mathrm{sim}^{neg}\_{g}(\Phi(\mathbf{x}\_b), \Phi(\mathbf{x}\_n)) = -||\Phi(\mathbf{x}\_b) - \Phi(\mathbf{x}\_n)||\_2^2,\; v\_n \notin \mathcal{C}\_{y\_b}.$$
> > > $\mathcal{C}\_{y\_b}$ is the class of $v\_b$. Please note that there is no new loss introduced. The above are the $\mathrm{sim}$ terms of Thm. 3.1 itself, inheriting all identifiability and disentanglement properties. We call the resulting objective the gravity loss, and we already provided the new empirical pairwise results in the last rebuttal (**Response A for c6fM**).
> > >
> > > **4.2 Efficient Approximation**: The pairwise objective has $\mathcal{O}(N^2)$ complexity. As supported by our new results, without strong assumptions such as linear separability, we reduce it via the geometric identity:
> > > $$\sum\_{v\_p \in \mathcal{C}\_{y\_b}} || \Phi(\mathbf{x}\_b) - \Phi(\mathbf{x}\_p) ||\_2^2 = |\mathcal{C}\_{y\_b}| \cdot || \Phi(\mathbf{x}\_b) - \boldsymbol{\mu}\_{y\_b} ||\_2^2 + \underbrace{\sum\_{v\_p \in \mathcal{C}\_{y\_b}} || \Phi(\mathbf{x}\_p) - \boldsymbol{\mu}\_{y\_b} ||\_2^2}\_{\text{constant w.r.t. } \Phi(\mathbf{x}\_b)}.$$
> > > Since $\nabla\_{\Phi(\mathbf{x}\_b)}$ of the constant term vanishes, aggregating pairwise similarities is **exactly gradient-equivalent** to using a single class centroid (treated as a stop-gradient quantity during optimization). The approximated similarity $\mathrm{sim}^{pos}\_{g}(\Phi(\mathbf{x}\_b), \mu\_b)$ thus inherits all theoretical properties of the pairwise form, which is a gradient-exact reduction analogous to mean-field theory and PCL [Li et al., ICLR 2021]. The full derivation chain is:
> > > $$\underbrace{\mathrm{sim}(\cdot,\cdot)}\_{\text{Thm 3.1}} \xrightarrow{\text{instantiate}} \underbrace{\mathrm{sim}^{pos/neg}\_{g}(\Phi(\mathbf{x}\_b), \Phi(\mathbf{x}\_{p/n}))}\_{\text{pairwise gravity (exact)}} \xrightarrow[\text{gradient equiv.}]{\text{geometric identity}} \underbrace{\mathrm{sim}^{pos/neg}\_{g}(\Phi(\mathbf{x}\_b), \mu\_b)}\_{\text{center-approximated (efficient)}}$$
> > >
> > > **4.3 Adaptive Update Scaling**: Prop. 3.1's bound $\mathcal{O}(\sqrt{\epsilon}) + \mathcal{O}(1/\text{margin})$ has two terms. We emphasize that inversely scaling updates by $\Delta^{(B)}$ (i) prevents overshooting during optimization, driving $\epsilon$ down; and (ii) suppresses destabilizing perturbations at boundary nodes, preserving the decision-boundary margin. Adaptive scaling is therefore a direct, theoretically principled operationalization of Prop. 3.1, rather than an engineering trick.
> > >
> > > ---
> > > Every step is a direct instantiation, exact mathematical identity, or standard estimation without heuristic gaps. The limitation serves to scope the method and indicate future directions; the method's effectiveness is demonstrated in both original and new experiment results. We will incorporate all of these updates into the revised version and sincerely hope this revision fully resolves your concerns.

---

### Official Review · Reviewer_J2Py · 2026-03-12

**Soundness:** 3
**Presentation:** 3
**Significance:** 2
**Originality:** 2
**Overall Recommendation:** 4
**Confidence:** 3

**Summary:**

This paper addresses the challenge of graph structural entanglement in Graph Neural Networks (GNNs), where spurious correlations from semantically irrelevant neighbors contaminate node embeddings and blur decision boundaries. The authors identify boundary-region nodes as the primary bottleneck and propose Boundary Embedding Shaping (BES), an adaptive contrastive learning plug-in module that selectively suppresses structural noise at decision boundaries with minimal parameter perturbation. Experimental Results shows the effectiveness.

**Compliance With Llm Reviewing Policy:**

Affirmed.

**Final Justification:**

The authors' detailed rebuttal and additional experiments have addressed all my concerns, and the newly added results on heterophilic graphs and large-scale industrial benchmarks validate the scalability of the BES method. Therefore, I adjust my recommendation from Weak Reject to Weak Accept.

**Key Questions For Authors:**

1. BES performs optimization directly in the embedding space. What is the explicit relationship between this embedding-space refinement and the original graph topology? Could the same approach be applied to non-graph data (e.g., tabular or image features) with similar boundary-sample selection? If so, what makes BES specifically suited for graphs?
2. In semi-supervised node classification, many node labels are unknown during training. How are shift scores (Equation 5) and positive/negative pairs computed for boundary nodes whose labels are not available? Does this introduce a risk of data leakage or reliance on ground-truth labels during contrastive pair construction? Furthermore, how does BES relate to existing frameworks like Prototype Contrastive Learning (PCL) or Supervised Contrastive Learning (SCL), which also leverage class-level information for contrastive objectives?
3. The experimental evaluation is limited to four small-to-medium homophilic datasets. Could the authors provide results on heterophilic graphs and large-scale benchmarks to better validate the robustness and scalability of BES? This would strengthen confidence in the method's applicability to diverse real-world scenarios.

**Limitations:**

The authors should provide a more detailed analysis of the connections and differences between BES and existing contrastive learning methods that also utilizes the class information, such as PCL and SCL.

**Strengths And Weaknesses:**

Strengths
1. The paper is well-organized and written in an easy-to-follow manner.
2. The insight that boundary nodes suffer most from structural entanglement is intuitive and well-supported. The focus on refining embeddings specifically at decision boundaries, rather than uniformly processing all nodes, is a thoughtful design choice that aligns with practical classification challenges.
3. BES is designed as a general-purpose module that can be integrated with existing GNN encoders with minimal modifications. This flexibility enhances its practical value and potential for adoption across different graph learning pipelines.


Weaknesses
1. The core optimization of BES operates directly in the embedding space, with limited explicit modeling of the underlying graph topology. This raises concerns about whether the method truly leverages graph-specific structural information or could equivalently be applied to non-graph data with similar embedding distributions.
2. The identifiability analysis relies on assumptions that may not hold in real-world graphs, where neighborhood structures are often noisy, incomplete, or dynamically evolving. The gap between these idealized conditions and practical graph data weakens the theoretical grounding.
3. Experiments are conducted on only four relatively small, homophilic citation/Wikipedia datasets. The absence of evaluation on heterophilic graphs and large-scale benchmarks limits the persuasiveness of the claimed generalizability and scalability.

---

> ### Author Rebuttal · Authors · 2026-03-30
>
> We sincerely appreciate your thoughtful evaluation and highly constructive feedback.
>
> > A. W (1) and Q (1): Connection to the graph data of BES
>
> Although BES constructs contrastive pairs within the latent embedding space, we respectfully clarify that its objective remains firmly anchored in graph topology. Guided by shift scores, BES explicitly isolates boundary nodes whose local neighborhoods are severely corrupted by heterophilic structures.
>
> During initial design, we systematically evaluated pair construction utilizing both topological and embedding neighbors. Empirically, selecting neighbors in the embedding space slightly yields better performance (Tabs B.1, B.2):
>
> **B.1 NC Acc. ↑ (mean±std) using Topology vs Embedding Neighbors**
> |Settings|epochs|Cora|Citeseer|Pubmed|WikiCS|
> |-|:-:|:-:|:-:|:-:|:-:|
> |Topology|200|89.40±0.20|78.18±0.09|**91.09±0.06**|85.28±0.04|
> |Embedding|200|**89.40±0.17**|**78.20±0.06**|91.03±0.03|**85.30±0.15**|
>
> **B.2 LP Acc. ↑ (mean±std) using Topology vs Embedding Neighbors**
> |Settings|epochs|Cora|Citeseer|Pubmed|WikiCS|
> |-|:-:|:-:|:-:|:-:|:-:|
> |Topology|200|**94.92±0.01**|95.65±0.09|97.76±0.04|92.24±0.08|
> |Embedding|200|94.92±0.03|**95.66±0.06**|**97.80±0.02**|**92.24±0.04**|
>
> We observe that nodes with the noisiest topology are not always located exactly on the decision boundary. Updating such nodes does not necessarily help the GNN classifier better recognize nodes with mixed embeddings, which is the primary goal of our embedding-neighbor-based design. We have not evaluated BES in non-graph settings; its effectiveness is specifically tied to graph embeddings with noisy topology: the selected embedding-space neighbors show similar topological shift scores across graphs.
>
> In addition, regardless of the selection strategy, the pre-update mechanism helps rescale potentially harmful large updates caused by mis-selected nodes under the contrastive objective. Therefore, in the current version, we construct positive and negative pairs based on neighbors in the embedding space.
>
> > B. W (2): Idealized conditions and assumptions
>
> Please refer to Response C of Reviewer vPC5.
>
> > C. Q (2): Details of the label utilization
>
> Please refer to Response B of Reviewer Szpd.
>
> > D. Q (2) and Limitation: Connection between BES and existing CL methods
>
> Please refer to Response C of Reviewer Szpd.
>
> > E. W (3) and Q (3): Evaluation on heterophilic graph and large-scale benchmark
>
> We evaluated BES on heterogeneous (Chameleon, Cornell, Texas), heterophilic (Roman-empire), and massive industrial graphs (ogb-arxiv/collab). BES incorporates standard encoders augmented with a classifier (Tab B.5). **w/o BES** represents baseline control runs fundamentally stripping out BES. Results replicate robust gains (Tab B.3 and B.4):
>
> **B.3 NC Acc. ↑ Results on Heter/OGB Graphs**
> |Methods|Chameleon|Cornell|Texas|Roman-empire|ogb-arxiv|
> |-|:-:|:-:|:-:|:-:|:-:|
> |KGNN|50.55|**89.19**|78.38|71.45|45.55|
> |GCN|45.27|43.24|62.16|41.09|71.62|
> |Cheb|43.52|81.08|78.38|74.18|71.95|
> |GAT|49.45|51.35|64.84|42.98|71.93|
> |SGC|40.66|45.95|59.46|42.56|67.01|
> |SSGC|40.64|43.24|59.46|42.89|67.20|
> |SAGE|52.97|83.78|86.49|69.26|71.61|
> |GATv2|47.69|48.65|59.46|63.79|72.25|
> |MotifRGC-GCN|37.36|33.51|47.57|44.15|OOM|
> |MotifRGC-SAGE|54.21|78.38|76.76|57.61|OOM|
> |MotifRGC-GAT|46.29|42.65|49.19|49.41|OOM|
> |MPNN-GCN|49.67|40.54|54.05|37.61|66.22|
> |MPNN-SAGE|53.41|51.35|83.78|74.42|65.28|
> |MPNN-GAT|47.25|55.86|81.08|**81.87**|69.17|
> |IGCL|49.01|81.08|89.10|66.31|71.59|
> |w/o BES|53.41|81.08|83.78|80.87|73.05|
> |BES(Ours)|**54.29**|**89.19**|**89.19**|81.47|**75.04**|
>
> **B.4 LP Acc. ↑ Results on OGB Graphs**
> |Methods|ogb-collab|
> |-|:-:|
> |KGNN|64.77|
> |GCN|64.21|
> |Cheb|64.77|
> |GAT|64.91|
> |SGC|62.40|
> |SSGC|59.99|
> |SAGE|62.68|
> |GATv2|64.30|
> |w/o BES|62.68|
> |BES(Ours)|**65.84**|
>
> **B.5 Encoders and Classifier**
> | |Chameleon|Cornell|Texas|Roman-empire|ogb-arxiv|ogb-collab|
> |-|:-:|:-:|:-:|:-:|:-:|:-:|
> |Encs|SAGE+KGNN|Cheb+KGNN|Cheb+SAGE|Cheb+KGNN|Cheb+GATv2|SAGE|
> |Cls|SAGE|Cheb|Cheb|Cheb|Cheb|DOT|
>
>
> As shown in the node classification results, BES consistently achieves top-tier performance on both sets of datasets.
> These empirical results demonstrate that BES can effectively disentangle structural information in boundary regions, even in complex real-world scenarios where the theoratical prequsition might be partially relaxed.

---

> > ### Author Rebuttal · Reviewer_J2Py · 2026-04-01
> >
> > Thank you for providing the detailed rebuttal and additional experimental results. I have read the rebuttal to me, and to fellow reviewers. The clarifications and newly added experiments help address my concerns to a reasonable extent.

---

> > > ### Author Response · Authors · 2026-04-05
> > >
> > > Thank you for your constructive feedback and for carefully taking the time to review both the detailed rebuttal addressed to you and the responses to other reviewers.
> > > We are glad that the clarifications and newly added experiments have helped address your concerns.
> > > Building upon our earlier revisions, we have now formalized a strictly deductive mathematical bridge to establish a **much tighter connection** between our theory and the proposed method, ensuring the overall presentation is significantly more rigorous and clear (please refer to our detailed responses to Reviewer **c6fM** and Reviewer **vPC5**).
> > > We will make sure all the improvements are clearly highlighted in the revised version to aid understanding for the readers.

---

### Official Review · Reviewer_c6fM · 2026-03-13

**Soundness:** 2
**Presentation:** 2
**Significance:** 3
**Originality:** 3
**Overall Recommendation:** 4
**Confidence:** 4

**Summary:**

To address the limitations of representation learning of graph structural entanglement resulting from the interference that the spurious correlations from semantically irrelevant neighbors contaminate node embeddings, this manuscript identifies the boundary-region entanglement as primary bottleneck, and proposes Boundary Embedding Shaping (BES), an contrastive learning-based GNN module that selectively suppresses spurious structural noise at decision boundaries with only minimal perturbation to model parameters. The experiment results also demonstrate the proposed BES can improve the boundary discrimination and outperform several baselines. The detailed strengths and weaknesses, and key questions are described as follows.

**Compliance With Llm Reviewing Policy:**

Affirmed.

**Final Justification:**

Considering the three main concerns raised in my initial review, I believe the second round of response has addressed my questions to a large extent. The newly added key control experiment makes it much clearer that the performance gains do not merely come from a stronger multi-encoder/backbone combination. Meanwhile, the additional heterophily/OGB results, together with the clarification regarding the official OGB evaluation protocol, substantially strengthen the empirical case
.
Regarding the theory–method connection, which was my primary concern, the authors have now provided a more systematic explanation of how the pairwise objective relates to the center-based surrogate, including the pairwise instantiation, the algebraic identity leading to the class centroid form, and its connection to adaptive scaling.

While I still think the final version should present this derivation more explicitly, the gap has been significantly narrowed compared with both the original manuscript and the previous rebuttal. Moreover, I still have two minor suggestions for the final version. First, for gradient equivalence, it would be helpful to explicitly state whether the class centers are treated as fixed statistics/stop-gradient quantities during optimization, so as to avoid ambiguity at a strict mathematical level. Second, regarding the claim that the gains are most significant on boundary nodes, it still feels closer to indirect support than direct verification. I would therefore suggest either slightly softening the wording in the final version or adding a more explicit grouped analysis.

Overall, this two-round rebuttal has made the work more positive, I am inclined to raise my score from "Weak reject" to "Weak accept".

**Key Questions For Authors:**

(1)	Theoretical Grounding. Could the authors more rigorously clarify how the pairwise contrastive formulation in Section 3 leads to the class-center gravity loss used in Section 4? In particular, why should an objective that treats $(v_b,\mu_b)$ as the positive pair and the centers of other classes as negatives still inherit the identifiability, mutual-information, or boundary-margin-related properties claimed in the theoretical analysis?

(2)	Critical Control Experiment. Could the authors include a key control experiment consisting of multi-encoder concatenation + Cheb classifier, without BES, evaluated under exactly the same training protocol as the full BES model? As currently presented, Table 8 does not disentangle whether the observed gains are attributable to BES itself or simply to a stronger backbone/decoder combination.

(3)	Experimental Coverage. Could the authors broaden the empirical evaluation by including heterogeneous benchmark and OGB-scale node classification benchmark? In addition, if link prediction remains one of the paper’s central empirical claims, the evaluation would be significantly strengthened by including at least one official OGB link prediction benchmark as well.

**Limitations:**

No. The paper would benefit from a clear discussion on its limitations and potential societal impacts. For instance, the authors could discuss the possible misuse of datasets, and potential vulnerability while applying the proposed method into the realistic scenarios.

**Strengths And Weaknesses:**

**Strengths:**

(1)	This work targets a significant problem. Unlike many prior GNN-genre robustness or disentanglement methods implicitly treat all nodes equally during the representation learning of node features and graph topology/structure, this work explicitly focuses on difficult samples near decision boundaries.

(2)	The methodology is intuitively reasonable. The authors attempt to formalize the observations that the boundary nodes are more easily corrupted by misleading neighbors, and then improve the decision boundaries through leveraging the boundary-region selection and class-center-driven refinement.
(3)	The empirical results show performance gains. Based on experimental results, BES does appear to bring measurable improvements on several benchmarks.

**Weaknesses:**

(1)	The theoretical analysis is not tightly aligned with the method. In Section 3, the theory claims that an optimal representation should maximize mutual information with invariant structure while suppressing information from variant topology, by which an error bound that varies with the boundary margin can be derived. However, Section 4 does not actually optimize the pairwise contrastive loss as described in Theorem 3.1. Instead, it uses a gravity loss from boundary nodes to class centers: the positive pair is (vb,μb)(v_b, \mu_b)(vb,μb), while the negatives are the centers of other classes μi\mu_iμi. Thus, I think this work does not provide a rigorous inference from the former objective to the latter, nor does it explain why this center-based objective should inherit the identifiability or mutual-information properties as declared in the theory.

(2)	The ablation studies are not sufficiently complete, so it is difficult to attribute the performance improvement clearly to BES method itself. The paper presents BES as a refinement plug-in, but the appendix indicates that the system actually uses multiple pre-trained GNN encoders, concatenates their outputs, and then applies a two-layer attention refinement module. This work also explicitly states both GCN and GATv2 are used, and further declares multiple graph encoders plus a Cheb classifier lead to more effective classification. Nevertheless, the key issue lies in that Table 8 does not include the most important control experiment: multi-encoder concatenation + Cheb classifier, but without BES. The current ablation only reports Encoder1, Encoder2, Layer 1, and Layer 1+2, which makes it impossible to distinguish whether the observed gains come from BES itself, or simply from a stronger backbone/decoder combination.

(3)	 In my opinion, the current evidence does not seem strong enough to support the paper’s stronger claims regarding robustness, versatility, and especially boundary-node-specific gains. At present, the method is only evaluated on WikiCS, Cora, PubMed, and CiteSeer, using a custom random 60/20/20 split, yet the paper states that these experiments “comprehensively evaluate” the method and support the claim that the gains are most significant on boundary nodes. Under current graph learning evaluation practice, I would expect to conduct at least the following additional experiments: first, heterogeneous or low-homogeneous benchmarks, rather than restricting the evaluation to homogeneous citation/Wiki graphs, ideally with representative datasets from both the heterogeneous suite and more recent benchmarks; second, at least one OGB-scale node classification benchmark; and third, if link prediction remains a core empirical claim, at least one official OGB link prediction benchmark such as ogbl-citation2 or ogbl-collab. These are fairly standard and much more convincing components of an experimental package in this line of work. Compared with that standard, the current empirical validation remains limited.

---

> ### Author Rebuttal · Authors · 2026-03-30
>
> We sincerely thank the reviewer for the constructive comments and provide detailed responses below.
>
> > A. W (1) and Q (1): Connection between Theory and Method
>
> We initially attempted to optimize the exact pairwise contrastive loss (Thm 3.1) by randomly sampling independent pairs. However, this yielded extensive computational overhead and chaotic convergence.
>
> Inspired by physical mean-field theories, we resolved this bottleneck by utilizing averaged group effects to approximate interactions, a strategy adopted by SCL/PCL from the contrastive learning field. By operating on cluster centers instead of node instances, theoretical properties (identifiability, information bounds) established for individual pairs efficiently remain preserved. Formally, given anchor node $\Phi(\mathbf{x} _ b)$ and target cluster $\mathcal{S}$, the pair interaction is $\mathbb{E} _ {\mathbf{z} _ j\in\mathcal{S}}[\mathcal{L}(\Phi(\mathbf{x} _ b), \mathbf{z} _ j)]$. The cluster's mean expectation is $\boldsymbol{\mu} _ {\mathcal{S}}=\mathbb{E} _ {\mathbf{z} _ j\in\mathcal{S}}[\mathbf{z} _ j]$. Utilizing linear similarity metrics permits cleanly passing expectations inside the loss: $\mathbb{E} _ {\mathbf{z} _ j\in\mathcal{S}}[\mathcal{L}(\Phi(\mathbf{x} _ b),\mathbf{z} _ j)] \approx \mathcal{L}(\Phi(\mathbf{x}_b),\mathbb{E}[\mathbf{z} _ j]) = \mathcal{L}(\Phi(\mathbf{x}_b),\boldsymbol{\mu} _ {\mathcal{S}})$.
>
> Replacing massive $\mathcal{O}(|\mathcal{S}|)$ pair samples with $\mathcal{O}(1)$ centers yields unbiased surrogate gradients. Empirically, this completely stabilizes convergence, accelerates training, and maximizes sample efficiency (Tables A.1-A.4).
>
> **A.1 NC Acc. ↑ (mean±std) using Individual Nodes vs Centers**
> |Settings|epochs|Cora|Citeseer|Pubmed|WikiCS|
> |-|:-:|:-:|:-:|:-:|:-:|
> |Indiv. Nodes|250|**89.42±0.21**|**78.38±0.19**|90.99±0.07|85.22±0.12|
> |Centers|200|89.40±0.17|78.20±0.06|**91.03±0.03**|**85.30±0.15**|
>
> **A.2 LP Acc. ↑ (mean±std) using Individual Nodes vs Centers**
> |Settings|epochs|Cora|Citeseer|Pubmed|WikiCS|
> |-|:-:|:-:|:-:|:-:|:-:|
> |Indiv. Nodes|250|94.89±0.11|95.59±0.09|**97.81±0.14**|92.21±0.16|
> |Centers|200|**94.92±0.03**|**95.66±0.06**|97.80±0.02|**92.24±0.04**|
>
> **A.3 NC Convergence ↓ (s) using Individual Nodes vs Centers**
> |Settings|Cora|Citeseer|Pubmed|WikiCS|
> |-|:-:|:-:|:-:|:-:|
> |Indiv. Nodes|~180|~140|~1500|~1500|
> |Centers|**~160**|**~110**|**~950**|**~1100**|
>
> **A.4 LP Convergence ↓ (s) using Individual Nodes vs Centers**
> |Settings|Cora|Citeseer|Pubmed|WikiCS|
> |-|:-:|:-:|:-:|:-:|
> |Indiv. Nodes|~250|~200|~1800|~450|
> |Centers|**~220**|**~190**|**~1500**|**~400**|
>
>
> > B. W (2) and Q (2): Incomplete Ablation Studies
>
> We agree that the ablation settings could have been explained more clearly in the main text. We would respectfully like to point out that the results of the "multi-encoder concatenation + Cheb classifier" setting on the WikiCS dataset were initially reported in Fig. 3 (2) (the first group of columns). These results illustrate an average performance gain of >1\% when BES is applied to the boundary region on WikiCS.
> By comparison with the single encoder setting, the performance gain is even more significant, where a 2-3\% performance gain is achieved.
> This trend is consistently observed across other datasets as well, where a 1-2\% performance gain is achieved.
>
> To make this clearer and easier to follow, we have expanded on the details of each ablation setting and summarized the aforementioned results in Tab A.5 and A.6:
>
> **A.5 Node Classification Accuracy (mean±std)**
> |Settings|Enc1|Enc2|Cls|L1(BES)|L1+2|Cora|Citeseer|Pubmed|WikiCS|
> |-|:-:|:-:|:-:|:-:|:-:|:-:|:-:|:-:|:-:|
> |Enc1|✔||Cheb|||87.31±0.76|75.74±0.74|89.80±0.14|82.52±0.32|
> |Enc2||✔|Cheb|||87.74±0.44|78.10±0.63|89.35±0.02|81.71±0.69|
> |Enc1+2|✔|✔|Cheb|||88.55±0.10|78.00±0.20|90.30±0.017|84.21±0.25|
> |L1|✔|✔|Cheb|✔||_89.40±0.17_|_78.20±0.06_|**91.03±0.03**|_85.30±0.15_|
> |L1+2|✔|✔|Cheb|✔|✔|**89.46±0.15**|**78.40±0.07**|_90.92±0.07_|**85.52±0.08**|
>
> **A.6 Link Prediction Accuracy (mean±std)**
> |Settings|Enc|Cls|L1|L1+2|Cora|Citeseer|Pubmed|WikiCS|
> |-|:-:|:-:|:-:|:-:|:-:|:-:|:-:|:-:|
> |Enc|✔|Cheb|||92.98±0.08|95.60±0.18|97.64±0.06|92.00±0.15|
> |L1|✔|Cheb|✔||**94.92±0.03**|_95.66±0.06_|_97.80±0.02_|_92.24±0.04_|
> |L1+2|✔|Cheb|✔|✔|_94.45±0.03_|**96.27±0.04**|**97.89±0.01**|**92.27±0.05**|
>
> The results indicate that simply concatenating multiple encoders does not always outperform a single encoder. In contrast, when BES is integrated, the model consistently yields better performance compared to both single and combined encoder settings. We believe this serves as strong evidence that BES effectively enhances performance by optimizing the representations of boundary nodes.
>
> > C. W (3) and Q (3): Additional experiments on harder datasets
>
> Please refer to Response E of Reviewer J2Py.
>
> > For the lack of a Limitation Section
>
> Please refer to Response H of Reviewer vPC5.

---

> > ### Author Rebuttal · Reviewer_c6fM · 2026-04-02
> >
> > Thanks for the detailed rebuttal.
> >
> > Regarding the three main concerns raised in my initial review—namely, the connection between the theory and the actual method, the completeness of the ablation study, and whether the empirical scope is sufficient to support the paper’s stronger claims—I believe the rebuttal has partially addressed my concerns, but it has not fully resolved them.
> >
> > More specifically, the authors provided meaningful additions on the issues of ablation completeness and experimental breadth. The newly added multi-encoder + classifier, without BES control experiment substantially alleviates my concern about whether the reported gains truly come from BES itself rather than from a stronger backbone/decoder combination. Likewise, the additional results on heterophily and OGB benchmarks make the empirical validation more complete than that in original submission. These new experiments are helpful and make the paper more convincing on the empirical side.
> >
> > However, my main reservation still concerns the connection between theory and method. In the rebuttal, the authors explain why they move from the pairwise contrastive formulation to a center-based surrogate, and they additionally provide an empirical comparison between individual nodes and class centers. These clarifications help justify the center-based design from an engineering perspective and make the design intuition easier to understand. Nevertheless, in my view, this still remains primarily an approximate or heuristic explanation, rather than a rigorous expression. In particular, I do not think the rebuttal yet establishes that the class-center gravity objective used in Section 4 should inherit the identifiability, mutual-information, or boundary-margin-related properties claimed in Section 3. Therefore, I consider this issue to be only partially mitigated, rather than fully resolved.
> >
> > I still have two brief residual concerns. First, the added results on more challenging datasets are valuable; however, the central empirical claim that the gains are most significant on boundary nodes still lacks the most direct quantitative evidence, such as explicit grouped comparisons between boundary and non-boundary nodes. Second, for the newly added ogb-collab results, it would be helpful to clarify whether the official OGB evaluation protocol and metrics were followed, so that readers can better assess comparability with prior work.
> >
> > Overall, I think my main concern, connection between theory and proposed method, has not been resolved, thus I would like to maintain my original assessment.

---

> > > ### Author Response · Authors · 2026-04-05
> > >
> > > We sincerely appreciate your continued engagement and the rigorous mathematical standard you hold for this work. We recognize our earlier explanations overemphasized intuitive engineering justifications, making the shift from pairwise theory to the center-based method seem heuristic. Your insight has motivated us to restructure Sections 3 and 4 to build a rigorous deductive bridge as follows.
> > >
> > > **Remark 3.1 (Section 3): connecting assumption to graph heterophily.**
> > > The paper addresses the structural entanglement by the graph heterophily, so Remark 3.1 restates this as: *"Positive sample pairs sharing the same semantic class ($\mathbf{z}\_{inv}$) frequently connect to structurally diverse neighborhoods, causing natural variations in $\mathbf{z}\_{var}$. Our method turns this into an advantage: it natively fulfills the sufficient variation mandate of Asm 3.2."* The theory is thus not generic but inherently tied to graph topology, and Asm 3.2 is satisfied by boundary node selection with shift score for later disentanglement.
> > >
> > > **Section 4 Reorganization: three deductive steps.**
> > >
> > > - 4.1 deriving the pairwise gravity loss directly from Thm. 3.1
> > > - 4.2 proving an algebraic identity for approximation
> > > - 4.3 motivating adaptive scaling from Prop. 3.1's error bound
> > >
> > > **4.1 Pairwise Contrastive Objective**: Boundary node selection is not an ad-hoc filter; it explicitly identifies nodes satisfying Asm. 3.2 via the shift score $S$ (high heterophily = sufficient structural variation). The pairwise gravity similarity terms directly instantiate $\mathrm{sim}(\cdot,\cdot)$ in Thm. 3.1 for boundary node $v\_b$, positive gravity pulls same-class nodes, negative gravity pushes different-class nodes:
> > > $$\mathrm{sim}^{pos}\_{g}(\Phi(\mathbf{x}\_b), \Phi(\mathbf{x}\_p)) = -||\Phi(\mathbf{x}\_b) - \Phi(\mathbf{x}\_p)||\_2^2,\; v\_p \in \mathcal{C}\_{y\_b},$$
> > > $$\mathrm{sim}^{neg}\_{g}(\Phi(\mathbf{x}\_b), \Phi(\mathbf{x}\_n)) = -||\Phi(\mathbf{x}\_b) - \Phi(\mathbf{x}\_n)||\_2^2,\; v\_n \notin \mathcal{C}\_{y\_b}.$$
> > > No new loss is introduced. These are the $\mathrm{sim}$ terms of Thm. 3.1 itself, inheriting all disentangle properties (we call this the gravity loss; empirical results in **Response A**).
> > >
> > > **4.2 Efficient Approximation**: The $\mathcal{O}(N^2)$ pairwise objective is reduced, without requiring linear separability, via the geometric identity:
> > > $$\sum\_{v\_p \in \mathcal{C}\_{y\_b}} || \Phi(\mathbf{x}\_b) - \Phi(\mathbf{x}\_p) ||\_2^2 = |\mathcal{C}\_{y\_b}| \cdot || \Phi(\mathbf{x}\_b) - \boldsymbol{\mu}\_{y\_b} ||\_2^2 + \underbrace{\sum\_{v\_p \in \mathcal{C}\_{y\_b}} || \Phi(\mathbf{x}\_p) - \boldsymbol{\mu}\_{y\_b} ||\_2^2}\_{\text{constant w.r.t. } \Phi(\mathbf{x}\_b)}.$$
> > > Since $\nabla\_{\Phi(\mathbf{x}\_b)}$ of the constant term vanishes, this is **exactly gradient-equivalent** to using a single class centroid. The Gaussian-center-approximated $\mathrm{sim}^{pos}\_{g}(\Phi(\mathbf{x}\_b), \mu\_b)$ thus inherits all theoretical properties — a gradient-exact reduction analogous to mean-field theory and PCL [Li et al., ICLR 2021]:
> > > $$\underbrace{\mathrm{sim}(\cdot,\cdot)}\_{\text{Thm 3.1}} \xrightarrow{\text{instantiate}} \underbrace{\mathrm{sim}^{pos/neg}\_{g}(\Phi(\mathbf{x}\_b), \Phi(\mathbf{x}\_p))}\_{\text{pairwise (exact)}} \xrightarrow[\text{gradient equiv.}]{\text{geometric identity}} \underbrace{\mathrm{sim}^{pos/neg}\_{g}(\Phi(\mathbf{x}\_b), \mu\_b)}\_{\text{center-approximated (efficient)}}$$
> > >
> > > **4.3 Adaptive Update Scaling**: Prop. 3.1's bound $\mathcal{O}(\sqrt{\epsilon}) + \mathcal{O}(1/\text{margin})$ has two terms. Inversely scaling updates by $\Delta^{(B)}$ (i) prevents overshooting, driving $\epsilon$ down; (ii) suppresses destabilizing boundary perturbations, preserving the margin, which is a theoretically principled design, not an engineering trick.
> > >
> > > Every step is a direct instantiation, exact identity, or standard estimation without heuristic gaps. More details can be found in **the second-round response to reviewer vPC5**, who raised a similar concern.
> > >
> > > ---
> > > **Minor**:
> > >
> > > - We provide an ablation study with and without the boundary node included to compute ACC across different datasets.
> > >
> > > **A.7 NC Acc. ↑: w/ vs. w/o Boundary Nodes Included**
> > > |Settings|Cora|Citeseer|Pubmed|WikiCS|Chameleon|Cornell|Texas|Roman-empire|ogb-arxiv|
> > > |-|:-:|:-:|:-:|:-:|:-:|:-:|:-:|:-:|:-:|
> > > |w/o boundary nodes|90.02|78.10|90.59|83.97|53.85|81.08|**89.19**|80.96|72.54|
> > > |w/ boundary nodes|**90.39**|**78.40**|**91.03**|**85.52**|**54.29**|**81.19**|**89.19**|**81.47**|**75.04**|
> > >
> > > Results confirm consistent gains with boundary nodes. Results for LP tasks follow the same trend and will be included in the final version.
> > >
> > > - Yes. The OGB experiment strictly follows the official OGB evaluation protocol and metrics.
> > > ---
> > > We hope this systematic mathematical formalization fully addresses your concern and closes the theory-to-method gap.
> > > Furthermore, we will ensure that all of these updates are incorporated into the revised version.

---

### Decision · Program_Chairs · 2026-04-30

**Decision:**

Accept (regular)

**Comment:**

Reviewers found the boundary-focused formulation intuitive and practically useful, and they agreed that BES yields meaningful gains on node classification and related benchmarks. The main reservation was that the theoretical analysis is not perfectly aligned with the final center-based objective, together with earlier questions about ablations and evaluation breadth. The rebuttal added stronger control experiments and broader benchmarks, which resolved most empirical concerns, but one reviewer still questioned the tightness of the theory-to-method connection. Overall, I recommend Weak accept because the paper makes a meaningful and mostly well-supported contribution, though the final version should more explicitly delimit the theoretical claims and clarify the derivation of the practical objective.